# Scalable Inference of Sparsely-changing Gaussian Markov Random Fields

**Salar Fattahi**
Department of Industrial & Operations Engineering
University of Michigan
Ann Arbor, MI 48109
`fattahi@umich.edu`

**Andrés Gómez**
Department of Industrial & System Engineering
University of Southern California
Los Angeles, CA 90089
`gomezand@usc.edu`

## Abstract

We study the problem of inferring time-varying Gaussian Markov random fields, where the underlying graphical model is both sparse and changes sparsely over time. Most of the existing methods for the inference of time-varying Markov random fields (MRFs) rely on the *regularized maximum likelihood estimation* (MLE), that typically suffer from weak statistical guarantees and high computational time. Instead, we introduce a new class of constrained optimization problems for the inference of sparsely-changing Gaussian MRFs (GMRFs). The proposed optimization problem is formulated based on the exact $\ell_0$ regularization, and can be solved in near-linear time and memory. Moreover, we show that the proposed estimator enjoys a provably small estimation error. We derive sharp statistical guarantees in the high-dimensional regime, showing that such problems can be learned with as few as one sample per time period. Our proposed method is extremely efficient in practice: it can accurately estimate sparsely-changing GMRFs with more than 500 million variables in less than one hour.

## 1   Introduction

Contemporary systems are comprised of massive numbers of interconnected components that interact according to a hierarchy of complex, unknown, and time-varying topologies. For example, with billions of neurons and hundreds of thousands of voxels, the human brain is considered as one of the most complex physiological networks [18, 22, 28, 30, 37]. The temporal behavior of today's interconnected systems can be captured via *time-varying Markov random fields (MRF)*. Time-varying MRFs are associated with a temporal sequence of undirected *Markov graphs* $\mathcal{G}_t(V, E_t)$, where $V$ and $E_t$ are the set of nodes and edges in the graph at time $t$. The node set $V$ represents the random variables in the model, while the edge set $E_t$ captures the conditional dependency between these variables at time $t$. A popular approach for the inference of MRFs is based on the *maximum-likelihood estimation* (MLE): to obtain a model based on which the observed data is most probable to occur [42].

Despite being known as theoretically powerful tools [20, 39], MLE-based methods suffer from several fundamental drawbacks which render them impractical in realistic settings. First, they often suffer from notoriously high computational cost in massive problems, where the number of variables to be inferred is in the order of millions, or more. Second, they struggle to incorporate sparsity amongst

their components, which is pervasive in large-scale systems. In particular, while *sparsely-changing* MRFs can in theory be accurately estimated using sparsity-promoting regularizers (such as $\ell_0$ penalty), most of the existing methods resort to relaxed or weaker variants of such regularization (such as $\ell_1$ penalty), thereby suffering from inferior statistical guarantees.

To address the aforementioned challenges, we propose a class of constrained optimization problems that achieve superior statistical and computational guarantees, compared to the regularized MLE, for the inference of time-varying Gaussian MRFs (GMRFs). Our approach departs from the usual wisdom in statistics and machine learning that inference problems with nonconvex $\ell_0$ terms are intractable, and convex proxies should be used instead. In particular, we show that the inference of sparsely-changing Gaussian MRFs can be solved efficiently via nonconvex $\ell_0$ penalties.

**Notations.** The $i^{th}$ element of a time-series vector $v_t$ is denoted as $v_{t;i}$; the $(i,j)^{th}$ element of a time-series matrix $V_t$ is denoted as $V_{t;ij}$. For a vector $v$, the notation $v_{i:j}$ is used to denote the subvector of $v$ from index $i$ to $j$. For a vector $v$, the notations $\|v\|_\infty, \|v\|_2, \|v\|_0$ denote the $\ell_\infty$ norm, $\ell_2$ norm, and the number of nonzero elements, respectively. Moreover, for a matrix $M$, the notations $\|M\|_2, \|M\|_\infty, \|M\|_{1/1}, \|M\|_{\infty/\infty}$ refer to the induced 2-norm, induced $\infty$-norm, $\ell_1/\ell_1$ norm, and $\ell_\infty/\ell_\infty$ norm, respectively. Moreover, we define $\|M\|_{\mathrm{off}} = \|M\|_{1/1} - \sum_{i=1}^d |M_{ii}|$. For a vector $v$ and matrix $M$, the notations $\mathrm{supp}(v)$ and $\mathrm{supp}(M)$ are defined as the sets of their nonzero elements. Given two sequences $f(n)$ and $g(n)$, the notation $f(n) \lesssim g(n)$ implies that there exists a constant $C < \infty$ that satisfies $f(n) \le Cg(n)$, and $f(n) \asymp g(n)$ implies that $f(n) \lesssim g(n)$ and $g(n) \lesssim f(n)$.

All proofs are deferred to the supplementary file.

## 1.1 Warm-up: Regularized MLE for Sparsely-changing GMRFs

Consider a multivariate zero-mean Gaussian process $\{X_t\}_{t=0}^T$ with distribution

$$\mathbb{P}(X_t) = \exp\left\{ -\frac{1}{2}\langle \Theta_t, X_t X_t^\top \rangle + \langle \eta_t, X_t \rangle - A(\mu_t, \Theta_t) \right\} \tag{1}$$

for $t = 0, \ldots, T$ where $A : \mathbb{R}^{d \times d} \to \mathbb{R}$ is the log-partition function used to normalize the distribution. Without loss of generality, we assume that the mean is zero. At any given time $t$, a sequence of data samples $\{X_t^{(i)}\}_{i=1}^{N_t}$ is collected from (1). The inference of time-varying GMRFs reduces to estimating the time-varying precision matrix $\Theta_t$ from the data samples $\{X_t^{(i)}\}_{i=1}^{N_t}$. Moreover, the edge set of the Markov graph $\mathcal{G}_t$ coincides with the off-diagonal nonzero elements of $\Theta_t$ [45].

We first illustrate the fundamental drawbacks of the $\ell_1$-regularized MLE for time-varying GMRFs with sparsely-changing structures. The sparse precision matrices can be estimated via the following regularized MLE, also known as *time-varying Graphical Lasso (GL)* [11, 16]:

$$\{\widehat{\Theta}_t\}_{t=0}^T = \arg\min_{\Theta_t} \ \sum_{t=0}^T \left( \langle \Theta_t, \widehat{\Sigma}_t \rangle - \log\det(\Theta_t) \right)$$

$$+ \gamma_1 \sum_{t=0}^T \|\Theta_t\|_{\mathrm{off}} + \gamma_2 \sum_{t=1}^T \|\Theta_t - \Theta_{t-1}\|_{1/1} \tag{2a}$$

$$\text{s.t.} \ \Theta_t \succ 0 \qquad t = 0, 1, \ldots, T \tag{2b}$$

where $\widehat{\Sigma}_t \in \mathbb{R}^{d \times d}$ is the sample covariance matrix at time $t$. Without loss of generality, we assume that the samples have zero mean. Example 1 below shows that (2) may lead to poor estimates.

**Example 1.** *Consider a scenario where $\{\Theta_t\}_{t=0}^4 \in \mathbb{R}^{25 \times 25}$ are randomly generated symmetric and sparse matrices. At each time $t = 0, \ldots, 4$, the precision matrix $\Theta_t$ has exactly 30 off-diagonal elements with value one in its upper-triangular part, and the remaining off-diagonal entries are set to zero. Moreover, the diagonal entries $\Theta_{t;ii}$ are chosen as $1 + \sum_{j \ne i} \Theta_{t;ij}$. At every time, 5 nonzero off-diagonal elements are changed to zero, and 5 zero elements are set to one. The sample covariance $\widehat{\Sigma}_t$ is obtained by collecting 500 samples from the Gaussian distribution with the constructed precision matrices. Figure 1a illustrates a heatmap of the mismatch error, i.e., the total number of mismatches in the sparsity patterns of the true and estimated precision matrices and their differences, for different values of the regularization coefficients. It can be seen that after an exhaustive search over the*

*regularization coefficient space, the best achievable mismatch error is in the order of 50. Thus, the estimated parameters reveal little information about the true structure of the time-varying GMRF.*

*Moreover, Figure 1b depicts the concatenation of the nonzero elements in the true precision matrices (dashed red line), and their corresponding values in the estimated matrices (blue curve). It can be seen that, even when the sparsity pattern of the elements is correctly recovered, the estimated nonzero entries are "shrunk" toward zero, incurring a substantial bias.*

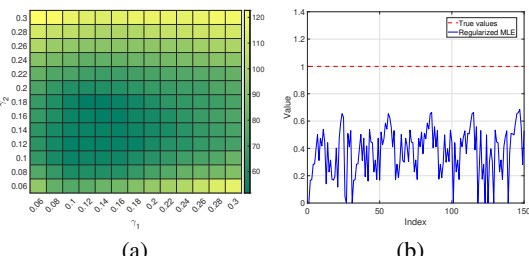

(a)                                  (b)

Figure 1: (a) The heatmap of the mismatch error. (b) The true and estimated nonzero elements of the precision matrix.

The above example shows the inferior statistical performance of the time-varying GL as an instance of a regularized MLE method for sparsely-changing GMRFs. In addition to its subpar statistical performance, time-varying GL suffers from expensive computational complexity: a general-purpose interior-point algorithm for solving time-varying GL has a prohibitive *per-iteration* complexity of $\mathcal{O}(Td^6)$ [10]. More recent algorithms for solving time-varying GL have lower *per-iteration* complexity of $\mathcal{O}(Td^3)$ [16, 26, 32]. However, these methods suffer from a slow (sublinear) convergence rate of $O(1/\epsilon)$, increasing the overall complexity to $\mathcal{O}(Td^3/\epsilon)$ in order to obtain an $\epsilon$-accurate solution. Thus, there is an inherent tradeoff between the quality of the solution found and the computational time required, with the performance deteriorating sharply as the number of precision digits increase. Solvers with such computational complexity may fall short of practical use in the large-scale settings. We now discuss the proposed method, which finds optimal solutions to the relevant optimization problems in *strongly polynomial time*.

## 2  Proposed Approach

The proposed framework is based on exact solutions to a class of tractable discrete $\ell_0$-problems, thus circumventing bias and other drawbacks of the standard $\ell_1$-approximations, while guaranteeing the scalability of the proposed method. As a general framework, we study the optimization problem:

$$\min\ (1-\gamma)\sum_{t=0}^{T}\|\Theta_t\|_0 + \gamma\sum_{t=1}^{T}\|\Theta_t - \Theta_{t-1}\|_0 \tag{3a}$$

$$\text{s.t. } \|\Theta_t - \widetilde{F}^*(\widehat{\Sigma}_t)\|_{\infty/\infty} \leq \lambda_t \qquad\qquad t = 0,1,\ldots,T \tag{3b}$$

$$\Theta_t \in \mathbb{R}^{d\times d} \qquad\qquad t = 0,1,\ldots,T \tag{3c}$$

where the optimal solutions $\{\widehat{\Theta}_t\}_{t=0}^{T}$ are the estimates of the precision matrices of the sparsely-changing GMRF, $\widehat{\Sigma}_t \in \mathbb{R}^{d\times d}$ is the sample covariance matrix at time $t$, and $\widetilde{F}^*(\cdot)$ is an *approximate backward mapping* of the model. In particular, we use the approximate backward mapping proposed in [46], see §4.1 for a formal definition.

First, we establish a deterministic guarantee on the estimation error of the optimal solution to (3).

**Theorem 1** (Estimation error and sparsistency). *Let $0 < \gamma < 1$. For every $t = 0,\ldots,T$, define $\mathcal{S}_t$ as the set of indices corresponding to the nonzero elements of the true precision matrix $\Theta_t^*$, and define $\mathcal{D}_t$ as the set of indices corresponding to the nonzero elements of $\Theta_t^* - \Theta_{t-1}^*$. Assume that*

- $\left\|\Theta_t^* - \widetilde{F}^*(\widehat{\Sigma}_t)\right\|_{\infty/\infty} < \lambda_t,\ \forall 0 \leq t \leq T,$
- $2\lambda_t \leq \min_{(i,j)\in\mathcal{S}_t} |\Theta_{t;ij}^*|,\ \forall 0 \leq t \leq T,$
- $2\lambda_t + 2\lambda_{t-1} \leq \min_{(i,j)\in\mathcal{D}_t} |\Theta_{t;ij}^* - \Theta_{t-1;ij}^*|,\ \forall 0 \leq t \leq T.$

*Then, the following statements hold for every $0 \leq t \leq T$:*

**Sparsistency** $\text{supp}\left(\widehat{\Theta}_t\right) = \text{supp}\left(\Theta_t^*\right)$ *and* $\text{supp}\left(\widehat{\Theta}_t - \widehat{\Theta}_{t-1}\right) = \text{supp}\left(\Theta_t^* - \Theta_{t-1}^*\right)$.

**Estimation error** $\|\widehat{\Theta}_t - \Theta_t^*\|_{\infty/\infty} \leq 2\lambda_t$ *and* $\|\widehat{\Theta}_t - \Theta_t^*\|_2 \leq 2\sqrt{|\mathcal{S}_t|}\lambda_t$.

Theorem 1 presents a set of conditions under which the proposed estimation method achieves sparsistency and small estimation error. The first condition entails that the true precision matrix $\Theta_t^*$ is a feasible solution to (3). The second and third conditions imply that there is a non-negligible gap between the zero and nonzero elements of the true parameters and their temporal changes. In §4 we present additional bounds specific to our choice of backward mapping.

**Theorem 2** (Computational complexity). *Given $\widetilde{F}^*(\widehat{\Sigma}_t)$, the optimization problem (3) can be solved to optimality in at most $\mathcal{O}((dT)^2)$ time and memory on a single thread.*

Theorem 2 shows that the optimization problem (3) can be solved efficiently and in *strongly polynomial* time despite its non-convex nature. As will be explained later, our choice of approximate backwards mapping [46] requires inverting $T + 1$ matrices, each with size $d \times d$, thereby increasing the overall complexity to $\mathcal{O}((dT)^2 + Td^3)$. If $T \leq d$, the complexity of the our method is dominated by that of the matrix inversion, which is unavoidable, even if the sample covariance matrix coincides with its true analog (since we still need to invert them to obtain the true precision matrices).

Our solution method for (3) relies on the *element-wise decomposability* of (3): we decompose (3) into smaller subproblems over different coordinates of $\{\Theta_t\}_{t=0}^T$. Then, we show that the optimal solution to each subproblem can be obtained by solving a shortest path problem on an auxiliary *weighted directed acyclic graph (DAG)*. The details of the solution method are presented in §5. Moreover, the proposed algorithm is easily parallelizable, leading to better runtimes in practice.

**Example 1 (continued).** *Figure 2 depicts the performance of the proposed method, compared to the regularized MLE with $\gamma_1 = 0.14$ and $\gamma_2 = 0.16$ (corresponding to the smallest mismatch error) for the instances generated in Example 1. The regularization parameter $\gamma$ in the objective function of (3) is set to 0.2. Moreover, for simplicity, we set $\lambda_0 = \cdots = \lambda_4 = \lambda$. Figure 2a demonstrates that the proposed method enjoys a significantly smaller mismatch error, for a wide range of $\lambda$. On the other hand, Figure 2b shows that the synthetic bias caused by the $\ell_1$ penalty in the regularized MLE is alleviated via the proposed method.*

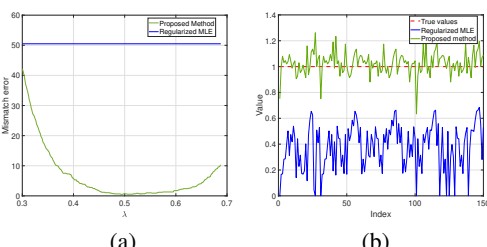

(a)  (b)

Figure 2: (a) The mismatch error of the proposed method for different values of $\lambda$ compared to the regularized MLE. (b) The true and estimated nonzero elements of the precision matrix.

## 3   Related Works

**Time-varying MRF.** In addition to the time-varying GL introduced in §1.1, a recent line of works have studied the inference of smoothly-changing GMRFs [23, 15, 48], where a kernel averaging technique combined with Graphical Lasso is used to estimate the smoothly-changing precision matrices. However, these methods do not leverage the prior information about the sparsity of the parameter differences. With the goal of addressing this deficiency, several works have studied the inference of sparsely-changing MRF (also known as sparse *differential networks*) [43, 47, 27]. However, the main drawback of these methods is that they only estimate the parameter differences, and their theoretical guarantees are restricted to problems with two time steps ($T = 0, 1$). Similarly, regression-based approaches have been proposed for change point detection problems [24, 36] with MRFs and two time periods, assuming the sparsity pattern of all entries of the precision matrices change at the same time. In contrast, [44] studies the inference of sparse MRFs given an index variable under the assumption that the sparsity pattern is invariant, whereas [14] assumes that the precision matrix is a linear function of the index variable.

**Sparsity-promoting optimization.** Optimization problems with $\ell_0$ terms are often deemed to be intractable, and approximations are solved instead. Perhaps the most popular approach is the *fused lasso* [34, 40, 38, 41], which calls for replacing terms $\|\Theta_t\|_0$ and $\|\Theta_t - \Theta_{t-1}\|_0$ with their $\ell_1$-approximations. Nonetheless, such approximations result in subpar statistical performance when compared with exact $\ell_0$ methods [19, 29].

Exact or near-optimal methods for optimization problems of the form

$$\min_{\theta \in [\ell, u]^p} \|\theta\|_0 + \sum_{i=1}^{p} \sum_{j=i+1}^{p} g_{ij}(\theta_i - \theta_j), \tag{4}$$

for given one-dimensional functions $g_{ij} : \mathbb{R} \to \mathbb{R}$, have also been studied in the literature. If functions $g_{ij}$ are convex, then problem (4) admits pseudo-polynomial time algorithm [3, 7]. Moreover, convex relaxations that deliver near-optimal solutions for (4) were proposed for the special case of convex quadratic $g$ functions [6]; if, additionally, we have $\ell = 0$ and $u = \infty$, then problem (4) is in fact solvable in strongly polynomial time [5]. On the other hand, problem (4) is much more challenging for non-convex $g$: if $g(x) = \mathbb{1}\{x \neq 0\}$, as is the case in (3), then problem (4) is NP-hard even if the term $\|\theta\|_0$ is dropped from the objective [17]. Nonetheless, as we show in this paper, problem (4) can be solved efficiently in the context of time-varying MRFs, where $g_{ij}(x) = 0$ whenever $j > i + 1$.

# 4  Statistical analysis

Theorem 1 presents a set of deterministic conditions under which the estimates from (3) enjoy zero mismatch error and small estimation error. However, the formulation of (3) is contingent upon the availability of an accurate backward mapping, and a choice of $\lambda_t$ that satisfies the conditions of Theorem 1. In this section, we show how to efficiently design sample-efficient approximate backward mappings, and select $\lambda_t$ accordingly for the class of sparsely-changing GMRFs. Moreover, we use the deterministic conditions of Theorem 1 to arrive at a non-asymptotic probabilistic guarantee for the inference of time-varying GMRFs under different prior knowledge on their temporal behavior, such as sparsity and smoothness.

## 4.1  Sparsely-changing GMRFs

Given the true covariance matrix $\Sigma_t$, the backward mapping of time-varying GMRFs as defined in (1) takes the form $F^*(\Sigma_t) = \Sigma_t^{-1}$. In light of this closed-form expression for the backward mapping, a commonly-used approximation is $\widetilde{F}^*(\widehat{\Sigma}_t) = \widehat{\Sigma}_t^{-1}$, where $\widehat{\Sigma}_t = \frac{1}{N_t} \sum_{i=1}^{N_t} X_t^{(i)} X_t^{(i)^\top}$ is the sample covariance matrix. However, in the high-dimensional settings where $d \gg N_t$, this approximate backward mapping is not well-defined, since the sample covariance matrix is highly rank-deficient.

To address this issue, [46] propose a *proxy* backward mapping for high-dimensional settings: consider the soft-thresholding operator $\mathtt{ST}_\nu(M) : \mathbb{R}^{d \times d} \to \mathbb{R}^{d \times d}$, where $\mathtt{ST}_\nu(M)_{ij} = M_{ij} - \mathrm{sign}(M_{ij}) \min\{|M_{ij}|, \nu\}$ if $i \neq j$, and $\mathtt{ST}_\nu(M)_{ij} = M_{ij}$ if $i = j$. The approximate backward mapping is then given by $\widetilde{F}^*(\widehat{\Sigma}_t) = [\mathtt{ST}_\nu(\widehat{\Sigma}_t)]^{-1}$, which is well-defined, even in the high-dimensional setting, with an appropriate choice of the threshold $\nu$ [46]. We make the following assumption.

**Assumption 1** (Bounded norm). *There exist constant numbers $\kappa_1 < \infty$, $\kappa_2 > 0$, and $\kappa_3 < \infty$ such that, for every $t = 0, \ldots, T$,*

$$\|\Theta_t\|_\infty \leq \kappa_1, \quad \inf_{w : \|w\|_\infty = 1} \|\Sigma_t w\|_\infty \geq \kappa_2, \quad \|\Sigma_t\|_{\infty/\infty} \leq \kappa_3.$$

Assumption 1 implies that the true covariance matrices and their inverses have bounded norms. Another key notion which plays an important role in our statistical guarantees is the *weak sparsity* of the covariance matrices, as defined next.

**Definition 1** (Weak sparsity). *Given $0 \leq q \leq 1$ and dimension $d$, define $s_d(q, d) \overset{\text{def}}{=} \max_i \sum_{j=1}^{d} |[\Sigma_t]_{ij}|^q$.*

Assuming that $s_d(0, d) \ll d$, Assumption 1 reduces to the covariance matrix being sparse. Moreover, in many cases, a sparse inverse covariance matrix leads to weakly sparse covariance matrices. For instance, if $\Theta_t$ has a banded structure with small bandwidth, then it is known that the elements of $\Sigma_t = \Theta_t^{-1}$ enjoy exponential decay away from the main diagonal elements [12, 21]. Under such circumstances, one can verify that $s_d(q, d) \leq \frac{C}{1 - \rho^q}$ for some constant $C > 0$ and $\rho < 1$. More generally, a similar statement holds for a class of inverse covariance matrices whose support graphs have large average path length [8, 9]; a large class of inverse covariance matrices with row- and column-sparse structures satisfy this condition. Theorem 3 states that the proposed method results in

high-quality solutions provided that the number of samples $N_t$ is sufficiently large with respect to the weak sparsity of the covariance matrices.

**Theorem 3.** *Consider a sparsely-changing GMRF and let $\zeta = \max\{\log_d(T+1), 1\}$. Given an arbitrary $\tau > \zeta + 2$, let $N_t \gtrsim s_d(q,d)^{\frac{2}{1-q}} \tau \log d$, for some $0 \le q < 1$. Then, with the approximate backward mapping $\widetilde{F}^*(\widehat{\Sigma}_t) = [ST_{\nu_t}(\widehat{\Sigma}_t)]^{-1}$, and parameters $\nu_t \asymp \sqrt{\frac{\tau \log d}{N_t}}$ and $\lambda_t \asymp \sqrt{\frac{\tau \log d}{N_t}}$, the estimates $\{\widehat{\Theta}_t\}_{t=0}^T$ obtained from (3) satisfy the following statements for all $t = 0, 1, \ldots, T$, with probability of at least $1 - 4d^{-\tau+\zeta+2}$:*

**Sparsistency** *We have* $\operatorname{supp}\left(\widehat{\Theta}_t\right) = \operatorname{supp}\left(\Theta_t^*\right)$ *and* $\operatorname{supp}\left(\widehat{\Theta}_t - \widehat{\Theta}_{t-1}\right) = \operatorname{supp}\left(\Theta_t^* - \Theta_{t-1}^*\right)$.

**Estimation error** $\|\widehat{\Theta}_t - \Theta_t^*\|_{\infty/\infty} \lesssim \sqrt{\frac{\tau \log d}{N_t}}$ *and* $\|\widehat{\Theta}_t - \Theta_t^*\|_F \lesssim \sqrt{\frac{\tau |S_t| \log d}{N_t}}$.

Theorem 3 is a direct consequence of Theorem 1 and provides, to the best of our knowledge, the first non-asymptotic guarantee on the inference of sparsely-changing GMRFs with an arbitrary length of time horizon $T$. In particular, it shows that the proposed optimization (3) guarantees small estimation error and zero mismatch error for sparsely-changing GMRFs, provided that $N_t$ scales logarithmically with the dimension of the precision matrices. In the static setting ($T = 0$), the derived bound recovers the existing results on the sample complexity of learning static GMRFs [25, 33, 35].

## 4.2 Sparsely-and-smoothly-changing GMRFs

In many applications, such as financial markets and motion detection in video frames, the associated graphical model should be learned "on-the-go", as the data arrives with a continuously changing graphical model. Under such circumstances, one may have access to few (or even one) samples at each time.

Suppose that the precision matrices change smoothly over time. Such smooth changes can be modeled via a continuous function $\Theta(x) : [0,1] \to \mathbb{R}^{d \times d}$ with uniformly bounded element-wise second derivatives $[\Theta(x)_{ij}]'' = \frac{d^2 \Theta(t)_{ij}}{dx^2}$, such that $\Theta_t^* = \Theta(t/T)$ [15, 48]. If $\Theta(t) \succeq aI$ for every $t \in [0,1]$ and some $a > 0$, then the covariance matrix $\Sigma(t) = \Theta(t)^{-1}$ is well-defined and smooth. Then, the problem of inferring the time-varying GMRF reduces to estimating a sequence of precision matrices $\{\Theta(0), \Theta(1/T), \ldots, \Theta(1)\}$ given the samples $X_t \sim \mathcal{N}(0, \Sigma(t/T))$. To alleviate the scarcity of samples, [15, 48] leverage the smoothness of the precision matrices, by taking the weighted average of the samples over time, where the weights are obtained from a nonparametric kernel. In particular, consider the weighted sample covariance matrix $\widehat{\Sigma}_t^w$:

$$\widehat{\Sigma}_t^w = \sum_{s=0}^t w(s,t) \Sigma_s, \text{ where } w(s,t) = \frac{1}{Th} K\left(\frac{s-t}{Th}\right) \tag{5}$$

and $K(\cdot)$ is a symmetric nonnegative kernel that satisfies a set of mild conditions which hold for most standard kernels, including the (truncated) Gaussian kernel. These conditions are delineated in the appendix. Next, we present the counterparts of Assumption 1 and Definition 1 for sparsely-and-smoothly-changing GMRFs.

**Assumption 2** (Bounded norm). *There exist constant numbers $\kappa_1 < \infty$, $\kappa_2 > 0$, and $\kappa_3 < \infty$ such that, for every $t \in [0, T]$,*

$$\|\Theta(t/T)\|_\infty \le \kappa_1, \quad \inf_{w:\|w\|_\infty=1} \|\Sigma(t/T)w\|_\infty \ge \kappa_2, \quad \|\Sigma(t/T)\|_{\infty/\infty} \le \kappa_3.$$

**Definition 2** (Weak sparsity). *Given $0 \le q \le 1$ and dimension $d$, define $s_c(q,d) \overset{def}{=} \max_{i,t\in[0,T]} \sum_{j=1}^d |[\Sigma(t/T)]_{ij}|^q$.*

We now present the analog of Theorem 3 for sparsely-and-smoothly-changing GMRFs.

**Theorem 4.** *Consider a sparsely-and-smoothly-changing GMRF with one sample per time, let $\zeta = \max\{\log_d(T+1), 1\}$, and suppose that the sample covariance matrices are constructed according to (5) with $h \asymp T^{-1/3}$. Given an arbitrary $\tau > \zeta + 2$, let $T \gtrsim s_c(q,d)^{\frac{3}{1-q}} (\tau \log d)^{3/2}$. Then, with the approximate backward mapping $\widetilde{F}^*(\widehat{\Sigma}_t) = ST_{\nu_t}(\widehat{\Sigma}_t^w)$ and parameters $\nu_t \asymp \frac{\sqrt{\tau \log d}}{T^{1/3}}$ and $\lambda_t \asymp \frac{\sqrt{\tau \log d}}{T^{1/3}}$, the estimates $\{\widehat{\Theta}_t\}_{t=0}^T$ obtained from (3) satisfy the following statements for all $T = 0, 1, \ldots, T$ with probability of at least $1 - d^{-\tau+\zeta+2}$:*

**Sparsistency** $\mathrm{supp}\left(\widehat{\Theta}_t\right) = \mathrm{supp}\left(\Theta_t^*\right)$ *and* $\mathrm{supp}\left(\widehat{\Theta}_t - \widehat{\Theta}_{t-1}\right) = \mathrm{supp}\left(\Theta_t^* - \Theta_{t-1}^*\right)$.

**Estimation error** $\|\widehat{\Theta}_t - \Theta_t^*\|_{\infty/\infty} \lesssim \sqrt{\frac{\tau \log d}{T^{2/3}}}$ *and* $\|\widehat{\Theta}_t - \Theta_t^*\|_F \lesssim \sqrt{\frac{\tau |\mathcal{S}_t| \log d}{T^{2/3}}}$.

Theorem 4 shows how the smoothness assumption on the true covariance matrix can be used to construct the backward mappings using the samples collected during the *entire* time horizon, thereby significantly reducing the sample complexity of learning time-varying GMRFs. In particular, leveraging the smoothness of the covariance matrix can reduce the minimum required number of samples from $\mathcal{O}(T \log d)$ to $\mathcal{O}((\log d)^{1.5})$. On the other hand, Theorem 3 does not impose any lower bound on $T$, and its estimation error decays faster in terms of the sample size.

## 5 Solution Method

In this section, we describe the proposed algorithm for solving (3). For the simplicity of notation, we define the lower bound and upper bound matrices $l_t$ and $u_t$ as $l_{t;ij} = [\widetilde{F}^*(\widehat{\Sigma}_t)]_{ij} - \lambda_t$ and $u_{t;ij} = [\widetilde{F}^*(\widehat{\mu}_t)]_{ij} + \lambda_t$, for every $1 \le i, j \le d$. The following fact plays a key role in our analysis.

**Fact 1.** *An optimal solution* $\left\{\widehat{\Theta}_t\right\}_{t=0}^T$ *of* (3) *satisfies for every* $1 \le i < j \le d$,

$$\left\{\widehat{\Theta}_{t;ij}\right\}_{t=0}^T \in \arg\min_{\{\Theta_{t;ij}\}_{t=0}^T} \ (1-\gamma)\sum_{t=0}^T \mathbb{1}\{\Theta_{t;ij} \ne 0\} + \gamma \sum_{t=1}^T \mathbb{1}\{\Theta_{t;ij} - \Theta_{t-1;ij} \ne 0\} \tag{6a}$$

$$\text{s.t.} \ \ l_{t;ij} \le \Theta_{t;ij} \le u_{t;ij} \quad \forall 0 \le t \le T. \tag{6b}$$

Fact 1 implies that (3) decomposes into the smaller subproblems (6). Therefore, our main focus is devoted to solving each subproblem independently. To further simplify the notation, we drop the subscript $ij$ from (6), whenever it is chosen arbitrarily, and use $\theta_t$ instead of $\Theta_{t;ij}$.

Let $\texttt{OPT}_{i \to j}(\gamma)$ denote the truncated problem from time $i$ to $j$ with the regularization coefficient $\gamma$:

$$f_{i \to j}^* = \min_{\{\theta_t\}_{t=i}^j} \ (1-\gamma)\sum_{t=i}^j \mathbb{1}\{\theta_t \ne 0\} + \gamma \sum_{t=i+1}^j \mathbb{1}\{\theta_t - \theta_{t-1} \ne 0\} \tag{7a}$$

$$\text{subject to} \ l_t \le \theta_t \le u_t \quad \forall i \le t \le j. \tag{7b}$$

Let the objective function for a candidate solution $\theta$ be denoted as $f_{i \to j}(\theta)$; by convention, we let $f_{i \to j}(\theta) = 0$ whenever $j < i$. Moreover, the optimal objective value and the set of optimal solutions to $\texttt{OPT}_{i \to j}(\gamma)$ are respectively denoted as $f_{i \to j}^*$ and $\mathcal{X}_{i \to j}^*$. Similarly, $\widehat{\theta}_{i \to j} \in \mathcal{X}_{i \to j}^*$ is used to denote an optimal solution to $\texttt{OPT}_{i \to j}(\gamma)$. We omit the subscript $i \to j$ whenever $i = 0$ and $j = T$. The $t^{\text{th}}$ feasible interval is defined as $\Delta_t = [l_t, u_t]$. Accordingly, the notation $\Delta_{t \to s}^\cap$ refers to $\Delta_{t \to s}^\cap \overset{\text{def}}{=} \Delta_t \cap \Delta_{t+1} \cap \cdots \cap \Delta_s$.

### 5.1 Special case: $\gamma = 1$

As the first step, we consider the special case $\gamma = 1$, and provide an efficient algorithm (Algorithm 1) for solving $\texttt{OPT}_{0 \to T}(1)$, where the sparsity is only promoted on the parameter differences (and not on the individual parameters). As will be shown later, Algorithm 1 will be used as a subroutine in our proposed algorithm for the general case $0 < \gamma < 1$. At a high level, Algorithm 1 recursively performs the following operations: at any given time $\tau$, the algorithm looks into the future to find a nonempty interval that is feasible for the longest possible time $\delta$. Then, it sets the subvector $\theta_{\tau:\delta}$ to an arbitrarily chosen element from this nonempty interval.

**Proposition 1.** $Greedy(l, u, 0, T)$ *returns an optimal solution* $\{\theta_t^{Greedy}\}_{t=0}^T$ *to* $\texttt{OPT}_{0 \to T}(1)$. *Moreover, the truncated solution* $\{\theta_t^{Greedy}\}_{t=0}^j$ *is optimal for* $\texttt{OPT}_{0 \to j}(1)$.

### 5.2 General case: $0 < \gamma < 1$

Now, we present our main algorithm for the general case $0 < \gamma < 1$.

---

**Algorithm 1** $\texttt{Greedy}(l, u, \tau, T)$

---

1: **Output:** Solution $\theta_{\tau \to T}^{\texttt{Greedy}}$, the objective value $f_{\tau \to T}^{\texttt{Greedy}}$ to $\texttt{OPT}_{\tau \to T}(1)$, and the index set $\Gamma$ of maximal nonempty intervals

2: Find largest $\delta$ such that $\Delta_{\tau \to \delta}^{\cap} \neq 0$;
3: Set $\theta_{\tau : \delta}^{\texttt{Greedy}} = \eta$ for some $\eta \in \Delta_{\tau \to \delta}^{\cap}$;
4: **if** $\delta \leq T - 1$ **then**
5:      Execute $\texttt{Greedy}(l, u, \delta + 1, T)$;
6: **end if**
7: Set $f_{\tau \to T}^{\texttt{Greedy}} = \sum_{t=\tau+1}^{T} \mathbb{1}\{\theta_t^{\texttt{Greedy}} - \theta_{t-1}^{\texttt{Greedy}} \neq 0\}$;
8: Set $\Gamma \leftarrow \Gamma \cup \{\delta\}$;
9: **Return** $\{\theta_t^{\texttt{Greedy}}\}_{t=\tau}^{T}$, $f_{\tau \to T}^{\texttt{Greedy}}$, and $\Gamma$;

---

**Definition 3.** *The set $\mathcal{Z}_{i \to j} = \{i, i+1, \dots, j\}$ is called a **zero-feasible sequence** if $l_k \leq 0 \leq u_k$ for every $k \in \mathcal{Z}_{i \to j}$. Moreover, the zero-feasible sequence $\mathcal{Z}_{i \to j}$ is called **maximal** if it is not strictly contained within another zero-feasible sequence.*

Let $\mathcal{Z}_{i_1 \to j_1}, \mathcal{Z}_{i_2 \to j_2}, \dots, \mathcal{Z}_{i_Z \to j_Z}$ be the set of all maximal zero-feasible sequences such that $0 \leq i_1 \leq j_1 < i_2 \leq j_2 < \cdots < i_Z \leq j_Z \leq T$, where $Z$ is the number of maximal zero-feasible sequences. If $Z = 0$, i.e., there is no zero-feasible sequence, then it is easy to see that $\sum_{t=0}^{T} \mathbb{1}\{\theta_t \neq 0\} = T + 1$ for every feasible solution, and hence, $\texttt{Greedy}(l, u, 0, T)$ leads to an optimal solution to (7). Another special case is when $i_1 = 0$ and $j_1 = T$, in which case the optimal solution is $\widehat{\theta} = 0$. Therefore, without loss of generality, suppose that $Z > 0$ and either $i_1 \neq 0$ or $j_1 \neq T$.

Our goal now is to obtain an optimal solution to (7) by solving a shortest path problem over a *weighted directed acyclic graph* (DAG) whose nodes correspond to the maximal zero-feasible sequences. In particular, consider a weighted DAG $\mathcal{G}$ with the vertex set $V = \{0, 1, \dots, Z, Z+1\}$, where the vertices $k$ and $l$ are connected via a directed arc $(k, l)$ if $k < l$. Moreover, for every arc $(k, l)$, the weight $W(k, l) = 0$ if $(k, l) = (0, 1), i_1 = 0$ or $(k, l) = (Z, Z+1), j_Z = T$, and

$$W(k, l) = (1 - \gamma)(i_l - j_k - 1) + \gamma f_{j_k+1 \to i_l-1}^{\texttt{Greedy}} + \gamma \mathbb{1}\{k \neq 0\} + \gamma \mathbb{1}\{l \neq Z+1\} \quad (8)$$

otherwise, where we define $j_0 = -1$ and $i_{Z+1} = T + 1$.

---

**Algorithm 2** Algorithm for solving (7)

---

1: **Output:** Optimal solution $\widehat{\theta}$ and the objective value $f^*$ to $\texttt{OPT}_{0 \to T}(\gamma)$
2: Find the maximal zero-feasible sequences;
3: Construct the DAG $\mathcal{G}$ with weights defined as (8);
4: Find the shortest path $p = (v_1, v_2, \dots, v_r)$ between the vertices $0$ and $Z + 1$ in $\mathcal{G}$;
5: Set $\widehat{\theta}_{j_{v_i}+1 : i_{v_{i+1}}-1} = \theta_{j_{v_i}+1 \to i_{v_{i+1}}-1}^{\texttt{Greedy}}$ and $\widehat{\theta}_{i_{v_l} : j_{v_{l+1}}} = 0$ for every $l = 1, 2, \dots, r$;
6: **Return** $\{\widehat{\theta}\}_{t=0}^{T}$ and $f^*$

---

**Theorem 5.** *The shortest path from $0$ to $Z + 1$ on $\mathcal{G}$ has value $f^*$.*

The above theorem implies that the optimal solution to (3) can be obtained via Algorithm 2.

**Theorem 6.** *Problem (7) can be solved in $\mathcal{O}(ZT)$ time and memory.*

Since $Z = \mathcal{O}(T)$ and a solution to (3) requires solving $\mathcal{O}(d^2)$ instances of (7), we find the total complexity stated in Theorem 2. Note however that if $Z = \mathcal{O}(1)$, then the overall complexity reduces to $\mathcal{O}(Td^2)$, which is linear in the total number of variables. In the next section, we will show that the practical runtime of the proposed algorithm is near-linear with respect to the number of variables.

## 6 Numerical Analysis

In this section, we evaluate the performance of the proposed estimator in synthetically generated massive datasets, and a case study on the correlation network inference in stock markets. We refer

the reader to the supplementary file for an extensive discussion of our simulations (e.g., the choice of parameters, robustness analysis, and a comparison with other state-of-the-art methods). In all of our simulations, the parameters $\nu_t$ and $\lambda_t$ are chosen directly from the data samples, i.e., without prior knowledge of the true solution, via Bayesian Inference Criterion (BIC) [31, 13].

**Case Study on Large Datasets**  We consider randomly generated instances of sparsely-changing GMRFs, where the true inverse covariance matrix is constructed as follows: at time $t = 0$, we set $\Theta_0 = I_d + \sum_{(i,j) \in \mathcal{S}} A^{(i,j)}$, where $A^{(i,j)}$ is a sparse positive semidefinite matrix with exactly two nonzero off-diagonal elements. For every $(i,j) \in \mathcal{S}$, we set $A_{ij}^{(i,j)} = A_{ji}^{(i,j)} = -0.4$ and $A_{ii}^{(i,j)} = A_{jj}^{(i,j)} = 0.4$. Clearly, $A^{(i,j)} \succeq 0$, and hence, $\Theta_0 \succ 0$. In the first experiment, we fix $T = 10$ and change the values of $d$. The number of nonzero elements in the individual precision matrices and their differences are set to $3d$ and $0.04d$, respectively. We evaluate the performance of the proposed method in the high dimensional settings, where $N_t = d/2$ for every $t = 0, \dots, T$. Moreover, define TPR and FPR for the individual parameters and their differences as the *true positive* and *false positive* values, normalized by the total number of nonzero and zero elements in the true precision matrices and their differences, respectively. Clearly, both TPR and FPR are between 0 and 1, with TPR $= 1$ and FPR $= 0$ corresponding to the perfect recovery of the sparsity patterns. Figure 7 depicts TPR, FPR, and the max-norm error of the estimated parameters, as well as the runtime of our algorithm for different values of $d$ with and without parallelization. It can be seen that both TPR and FPR values improve with the dimension for the estimated parameters and their differences. Moreover, with a single core, the runtime of our algorithm scales almost linearly with $d^2$, which is in line with the result of Theorem 2. Using 5 cores, the runtime of our algorithm is improved by 40% on average. Using 10 cores deteriorates the performance due to the shared memory limitations. *Using our algorithm, we reliably infer instances of sparsely-changing GMRFs with more almost 500 million variables in less than one hour.*

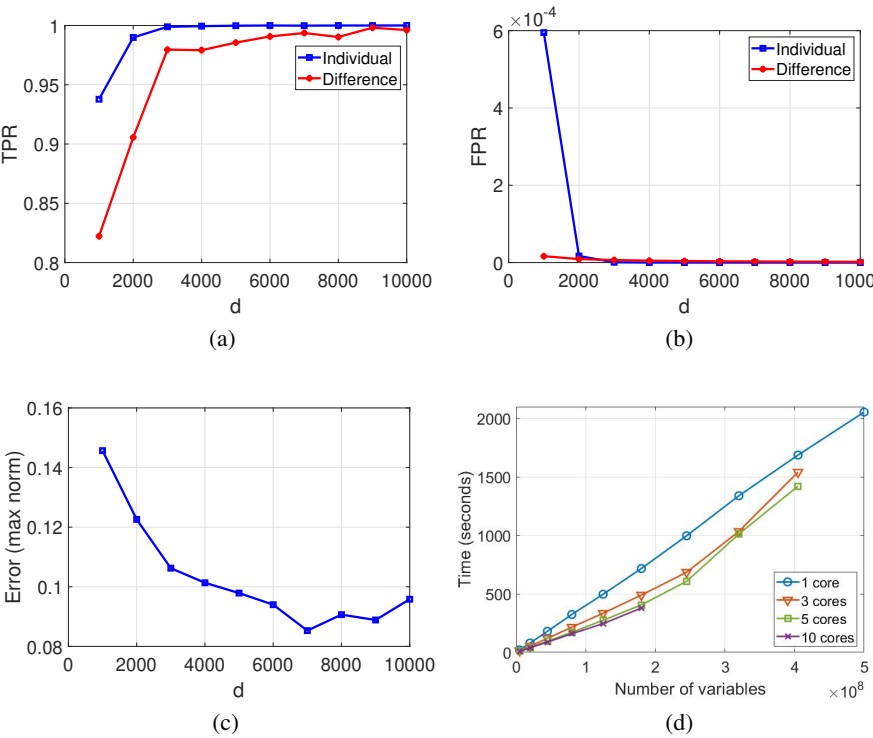

Figure 3: (a) TPR of the proposed method. (b) FPR of the proposed method. (c) The max-norm of estimation error. (d) The runtime of the proposed algorithm with and without parallelization.

**Case Study on Stock Market**  Finally, we illustrate the performance of our algorithm for the inference of stock correlation network. In particular, we consider an investor seeking to understand relationships between securities over time. Sparsity of the precision matrices guarantees interpretability,

while the sparsely-changing structure is imposed to identify sharp changes in market conditions, and a need to rebalance the portfolio. We consider the daily changes for 214 securities from 1990/01/04 to 2017/08/10, with the total number of 6990 days ($d = 214$ and $T = 6990$). Due to the continuously changing nature of the stock correlation network, we will use the kernel averaging approach introduced in Subsection 4.1. Using the constructed sample covariance matrices, we estimate the sparsely-changing precision matrix $\Theta(t/T)$ at discrete times $t \in \{30, 60, 90, \ldots, 6990\}$.

A drastic change in the correlation network signals a *spike* in the stock market, which may reflect the market's response to unexpected events. Figure 11 shows the number of changes in the estimated network, for the choices of $\nu_0 = 3$, $\lambda_0 = 0.16$, and $\gamma = 0.9$, together with the historical chart of National Association of Securities Dealers Automated Quotations (NASDAQ) [1]. It can be seen that the major spikes in the estimated network can be attributed to the historical stock market crashes. For instance, the spikes A, B, and C respectively correspond to the "early 1990s recession", "dot-com bubble", and "global financial crisis"; see [4] for more details. Interestingly, the estimated network can also detect other historical (but less severe) downturns in 2011 (point D) and 2016 (point E). In the supplementary materials, we provide a more detailed analysis with different choices of parameters.

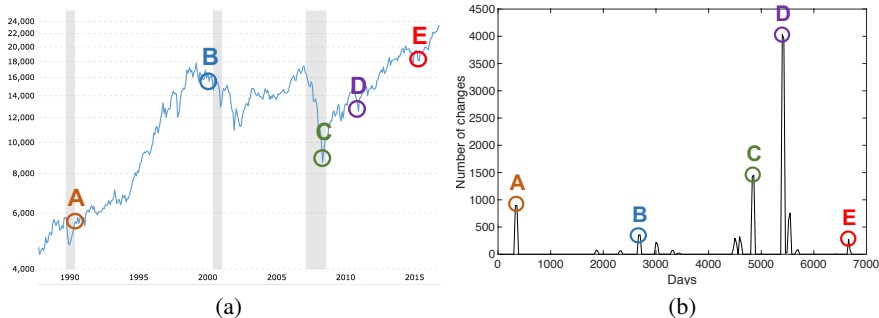

Figure 4: (a) NASDAQ historical chart [2]. (b) The number of changes in the estimated network.

## Acknowledgments and Disclosure of Funding

S.F. is supported by MICDE Catalyst Grant and MIDAS PODS Grant. A.G. is supported, in part, by the National Science Foundation under grant CIF 2006762.

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
