**A  Proofs**

 **A.1  Proof of Theorem 1**

508 Due to the feasibility of $\{\widehat{\Theta}_t\}_{t=0}^T$, one can write $\|\widehat{\Theta}_t - \widetilde{F}^*(\widehat{\Sigma}_t)\|_\infty \leq \lambda_t$. Combined with the first
509 assumption of the theorem, this implies that

$$
\begin{aligned}
\left\|\widehat{\Theta}_t - \Theta_t^*\right\|_\infty &= \left\|\widehat{\Theta}_t - \widetilde{F}^*(\widehat{\Sigma}_t) + \widetilde{F}^*(\widehat{\Sigma}_t) - \Theta_t^*\right\|_\infty \\
&\leq \left\|\widehat{\Theta}_t - \widetilde{F}^*(\widehat{\Sigma}_t)\right\|_\infty + \left\|\Theta_t^* - \widetilde{F}^*(\widehat{\Sigma}_t)\right\|_\infty \\
&< 2\lambda_t,
\end{aligned}
\tag{9}
$$

510 thereby establishing the element-wise estimation error bound. We proceed to show the sparsistency
511 of the estimated parameters. First, suppose that $\Theta_{t;ij}^* \neq 0$ for some time $t$ and index $(i,j)$. One can
512 write

$$
\begin{aligned}
\left|\widehat{\Theta}_{t;ij}\right| &= \left|\widehat{\Theta}_{t;ij} - \Theta_{t;ij}^* + \Theta_{t;ij}^*\right| \\
&\geq \left|\Theta_{t;ij}^*\right| - \left|\widehat{\Theta}_{t;ij} - \Theta_{t;ij}^*\right| \\
&> 0
\end{aligned}
\tag{10}
$$

513 where the last inequality is due to the second assumption of the theorem and (9). This implies that
514 $\operatorname{supp}(\Theta_t^*) \subseteq \operatorname{supp}(\widehat{\Theta}_t)$. Similarly, suppose that $\Theta_{t;ij}^* - \Theta_{t-1;ij}^* \neq 0$ for some time $t > 0$ and index
515 $(i,j)$. One can write

$$
\begin{aligned}
\left|\widehat{\Theta}_{t;ij} - \widehat{\Theta}_{t-1;ij}\right| &= \left|\widehat{\Theta}_{t;ij} - \Theta_{t;ij}^* + \Theta_{t;ij}^* - \Theta_{t-1;ij}^* + \Theta_{t-1;ij}^* - \widehat{\Theta}_{t-1;ij}\right| \\
&\geq \left|\Theta_{t;ij}^* - \Theta_{t-1;ij}^*\right| - \left|\widehat{\Theta}_{t;ij} - \Theta_{t;ij}^*\right| - \left|\widehat{\Theta}_{t-1;ij} - \Theta_{t-1;ij}^*\right| \\
&> 0
\end{aligned}
\tag{11}
$$

516 where the last inequality is due to the third assumption of the theorem and (9). This implies that
517 $\operatorname{supp}(\Theta_t^* - \Theta_{t-1}^*) \subseteq \operatorname{supp}(\widehat{\Theta}_t - \widehat{\Theta}_{t-1})$. Finally, due to the optimality of $\{\widehat{\Theta}_t\}_{t=0}^T$ and feasibility of
518 $\{\Theta_t^*\}_{t=0}^T$, one can write

$$
(1-\gamma)\sum_{t=0}^T \|\widehat{\Theta}_t\|_0 + \gamma\sum_{t=1}^T \|\widehat{\Theta}_t - \widehat{\Theta}_{t-1}\|_0 \leq (1-\gamma)\sum_{t=0}^T \|\Theta_t^*\|_0 + \gamma\sum_{t=1}^T \|\Theta_t^* - \Theta_{t-1}^*\|_0
$$

$$
\implies (1-\gamma)\sum_{t=0}^T \left( \sum_{(i,j)\notin\mathcal{S}_t} |\widehat{\Theta}_{t;ij}|_0 + \sum_{(i,j)\in\mathcal{S}_t} |\widehat{\Theta}_{t;ij}|_0 \right)
\tag{12}
$$

$$
+ \gamma\sum_{t=1}^T \left( \sum_{(i,j)\notin\mathcal{D}_t} |\widehat{\Theta}_{t;ij} - \widehat{\Theta}_{t-1;ij}|_0 + \sum_{(i,j)\in\mathcal{D}_t} |\widehat{\Theta}_{t;ij} - \widehat{\Theta}_{t-1;ij}|_0 \right)
$$

$$
\leq (1-\gamma)\sum_{t=0}^T \sum_{(i,j)\in\mathcal{S}_t} |\Theta_{t;ij}^*|_0 + \gamma\sum_{t=1}^T \sum_{(i,j)\in\mathcal{D}_t} |\Theta_{t;ij}^* - \Theta_{t-1;ij}^*|_0
$$

$$
\implies (1-\gamma)\sum_{t=0}^T \sum_{(i,j)\notin\mathcal{S}_t} |\widehat{\Theta}_{t;ij}|_0 + \gamma\sum_{t=1}^T \sum_{(i,j)\notin\mathcal{D}_t} |\widehat{\Theta}_{t;ij} - \widehat{\Theta}_{t-1;ij}|_0 \leq 0
\tag{13}
$$

519 where the last inequality follows from $\operatorname{supp}(\Theta_t^*) \subseteq \operatorname{supp}(\widehat{\Theta}_t)$ and $\operatorname{supp}(\Theta_t^* - \Theta_{t-1}^*) \subseteq \operatorname{supp}(\widehat{\Theta}_t - $
520 $\widehat{\Theta}_{t-1})$, which implies $\sum_{(i,j)\in\mathcal{S}_t} |\widehat{\Theta}_{t;ij}|_0 - |\Theta_{t;ij}^*|_0 \geq 0$ and $\sum_{(i,j)\in\mathcal{D}_t} |\widehat{\Theta}_{t;ij}^* - \widehat{\Theta}_{t-1;ij}|_0 - |\Theta_{t;ij}^* -$
521 $\Theta_{t-1;ij}^*|_0 \geq 0$ for every $t$. Due to $0 < \gamma < 1$, the above inequality implies that $\widehat{\Theta}_{t;ij} = 0$
522 for every $t$ and $(i,j) \notin \mathcal{S}_t$, and $\widehat{\Theta}_{t;ij} - \widehat{\Theta}_{t-1;ij} = 0$ for every $t > 0$ and $(i,j) \notin \mathcal{D}_t$. This
523 implies that $\operatorname{supp}(\widehat{\Theta}_t) \subseteq \operatorname{supp}(\Theta_t^*)$ and $\operatorname{supp}(\widehat{\Theta}_t - \widehat{\Theta}_{t-1}) \subseteq \operatorname{supp}(\Theta_t^* - \Theta_{t-1}^*)$. Finally, since
524 $\operatorname{supp}(\widehat{\Theta}_t) \subseteq \operatorname{supp}(\Theta_t^*)$, we have $|\operatorname{supp}(\widehat{\Theta}_t - \Theta_t^*)| = |\mathcal{S}_t|$. This, together with (9) implies that
525 $\|\widehat{\Theta}_t - \Theta_t^*\|_2 \leq \sqrt{|\mathcal{S}_t|}\|\widehat{\Theta}_t - \Theta_t^*\|_\infty \leq 2\sqrt{|\mathcal{S}_t|}\lambda_t$, thereby completing the proof. $\qquad\square$

## A.2 Proof of Theorem 3

For simplicity of notation, we drop the subscript from the definition of $s_d(\cdot, \cdot)$. The proof is inspired by Corollary 1 in [47]. First, we present the following key lemmas.

**Lemma 1** (Lemma 2 of [47] and Lemma 1 of [34]). *Suppose that $X^{(i)} \sim \mathcal{N}(0, \Sigma)$ for $i = 1, \ldots, N$, and $\widehat{\Sigma} = \frac{1}{N} \sum_{i=1}^{N} X^{(i)} X^{(i)^\top}$. Then, we have*

$$\left\| \widehat{\Sigma} - \Sigma \right\|_{\infty/\infty} \leq 8 \left( \max_i \Sigma_{ii} \right) \sqrt{\frac{\tau \log d}{N}} \tag{14}$$

*with probability of at least $1 - 4d^{-\tau+2}$ for any $\tau > 2$, provided that $N \geq 40 \left( \max_i \Sigma_{ii} \right)$.*

**Lemma 2** (Lemma 1 of [47]; modified). *Under the conditions of Lemma 1, we have*

$$\left\| ST_\nu(\widehat{\Sigma}) - \Sigma \right\|_\infty \leq 5\nu^{1-q} s(q,d) + 24\nu^{-q} s(q,d) \left( \max_i \Sigma_{ii} \right) \sqrt{\frac{\tau \log d}{N}} \tag{15}$$

*with probability of at least $1 - 4d^{-\tau+2}$ for any $\tau > 2$, provided that $N \geq 40 \left( \max_i \Sigma_{ii} \right)$.*

Based on the above lemmas, we proceed to present the proof of Corollary 3.

*Proof of Corollary 3.* It suffices to show that the conditions of Theorem 1 are satisfied with the proposed choices of $\lambda_t$ and $\nu_t$. It is easy to see that

$$\left\| \Theta_t - [\text{ST}_{\nu_t}(\widehat{\Sigma}_t)]^{-1} \right\|_{\infty/\infty} = \left\| [\text{ST}_{\nu_t}(\widehat{\Sigma}_t)]^{-1} (\text{ST}_{\nu_t}(\widehat{\Sigma}_t)\Theta_t - I) \right\|_{\infty/\infty}$$

$$\leq \left\| [\text{ST}_{\nu_t}(\widehat{\Sigma}_t)]^{-1} \right\|_\infty \left\| \Theta_t \right\|_\infty \left\| \text{ST}_{\nu_t}(\widehat{\Sigma}_t) - \Sigma_t \right\|_{\infty/\infty} \tag{16}$$

We provide separate bounds for different terms of the above inequality. Due to Assumption 1, one can write $\left\| \Theta_t \right\|_\infty \leq \kappa_1$. Moreover, due to Lemma 1, the following inequality holds with probability of at least $1 - 4d^{-\tau+2}$ for any $\tau > 2$

$$\left\| \text{ST}_{\nu_t}(\widehat{\Sigma}_t) - \Sigma_t \right\|_{\infty/\infty} \leq \left\| \text{ST}_{\nu_t}(\widehat{\Sigma}_t) - \widehat{\Sigma}_t \right\|_{\infty/\infty} + \left\| \widehat{\Sigma}_t - \Sigma_t \right\|_{\infty/\infty}$$

$$\leq \nu_t + 8\kappa_3 \sqrt{\frac{\tau \log d}{N_t}}$$

$$= 16\kappa_3 \sqrt{\frac{\tau \log d}{N_t}} \tag{17}$$

provided that $N_t \geq 40\kappa_3$ and $\nu_t = 8\kappa_3 \sqrt{\frac{\tau \log d}{N_t}}$. Finally, for any vector $w$, one can write

$$\left\| \text{ST}_{\nu_t}(\widehat{\Sigma}_t)w \right\|_\infty \geq \left\| \Sigma_t w \right\|_\infty - \left\| (\text{ST}_{\nu_t}(\widehat{\Sigma}_t) - \Sigma_t)w \right\|_\infty$$

$$\geq \left( \kappa_2 - \left\| \text{ST}_{\nu_t}(\widehat{\Sigma}_t) - \Sigma_t \right\|_\infty \right) \left\| w \right\|_\infty \tag{18}$$

On the other hand, the aforementioned choice of $\nu_t$ and Lemma 2 implies that

$$\left\| \text{ST}_{\nu_t}(\widehat{\Sigma}_t) - \Sigma_t \right\|_\infty \leq 64\kappa_3^{1-q} s(q,d) \left( \frac{\tau \log d}{N_t} \right)^{\frac{1-q}{2}} \tag{19}$$

Combining this inequality with (18) leads to

$$\left\| \text{ST}_{\nu_t}(\widehat{\Sigma}_t) - \Sigma_t \right\|_\infty \leq \frac{\kappa_2}{2} \tag{20}$$

provided that

$$N_t \geq \left( \frac{128 s(q,d)}{\kappa_2} \right)^{\frac{2}{1-q}} \kappa_3^2 \tau \log d \tag{21}$$

This implies that $\|\mathrm{ST}_{\nu_t}(\widehat{\Sigma}_t)w\|_\infty \geq \frac{\kappa_2}{2}\|w\|_\infty$, and hence, $\left\|[\mathrm{ST}_{\nu_t}(\widehat{\Sigma}_t)]^{-1}\right\|_\infty \leq \frac{2}{\kappa_2}$. Combining these bounds with (22) yields

$$\left\|\Theta_t - [\mathrm{ST}_{\nu_t}(\widehat{\Sigma}_t)]^{-1}\right\|_{\infty/\infty} \leq \frac{32\kappa_1\kappa_3}{\kappa_2}\sqrt{\frac{\tau\log d}{N_t}} = \lambda_t \tag{22}$$

with probability of at least $1 - 4d^{-\tau+2}$. Finally, we need to verify that the conditions $\lambda_t \leq \Theta_t^{\min}/2$ and $\lambda_t + \lambda_{t-1} \leq \Delta\Theta_t^{\min}/2$ hold. Based on the above definition of $\lambda_t$, it is easy to see that both of these conditions are satisfied if

$$N_t \geq \left(\frac{128\kappa_1\kappa_3}{\kappa_2}\right)^2 \max\left\{(\Theta_t^{\min})^{-2}, (\Delta\Theta_t^{\min})^{-2}, (\Delta\Theta_{t-1}^{\min})^{-2}\right\} \tau\log d$$
$$\implies N_t \gtrsim \tau\log d \tag{23}$$

Based on our assumption, we have $T + 1 \leq Cd^\zeta$ for some universal constant $C > 0$. Therefore, a simple union bound over $t = 0, \ldots, T$ implies that the statements of the corollary holds for every $t = 0, \ldots, T$ with the probability of at least

$$1 - 4\sum_{t=0}^{T} d^{-\tau+2} \geq 1 - 4(T+1)d^{-\tau+2} \geq 1 - 4d^{\zeta-\tau+2} \tag{24}$$

Selecting $\tau > \zeta + 2$ completes the proof. $\qquad\square$

## A.3   Proof of Theorem 4

First, we delineate the imposed assumptions on the selected kernel function.

**Assumption 3** ( [16]). *The kernel $K(x)$ satisfies the following conditions:*

- $\int_{-1}^{1} K(x)dx = 1$,

- $\int_{-1}^{1} x^2 K(x)dx \leq \infty$,

- $K(x)$ *is uniformly bounded on its support,*

- $\sup_{-1\leq x\leq 1} K''(x/h) = \mathcal{O}(h^{-4})$.

The following key lemmas are borrowed from [16].

**Lemma 3** (Lemma 5 of [16]). *For any fixed $t$, we have*

$$\|\mathbb{E}[\Sigma_t^w] - \Sigma(t/T)\|_{\infty/\infty} \lesssim C\left(h + \frac{1}{T^2 h^5}\right) \tag{25}$$

*for some constant $C > 0$.*

**Lemma 4** (Lemma 2 of [16]). *There exists a constant $c > 0$ such that*

$$\mathbb{P}\left(|[\Sigma_t^w]_{ij} - \mathbb{E}[\Sigma_t^w]_{ij}| \geq \epsilon\right) \leq 2\exp(-cTh\epsilon^2) \tag{26}$$

*for every $\epsilon > 0$ and any fixed $t$.*

Combining the above lemmas gives rise to the following result.

**Lemma 5.** *Assume that $h = T^{-1/3}$. Then, the following inequality holds for any $t$ and $\tau > 2$*

$$\left\|\widehat{\Sigma}_t^w - \Sigma(t/T)\right\|_{\infty/\infty} \lesssim \frac{\sqrt{\tau\log d}}{T^{1/3}} \tag{27}$$

*with probability of at least $1 - d^{-(\tau-2)}$.*

*Proof.* Based on Lemma 4, one can write

$$\mathbb{P}\left(\left\|\widehat{\Sigma}_t^w - \mathbb{E}[\Sigma_t^w]\right\|_{\infty/\infty} \geq \epsilon\right) \leq 2\exp(2\log d - cTh\epsilon^2) \tag{28}$$

569 Upon choosing $\epsilon = \sqrt{\frac{\tau \log d}{cTh}}$ for some $\tau > 2$, we have

$$\left\| \widehat{\Sigma}_t^w - \mathbb{E}[\Sigma_t^w] \right\|_{\infty/\infty} \leq \sqrt{\frac{\tau \log d}{cTh}} \tag{29}$$

570 with probability of at least $1 - d^{-(\tau-2)}$. Combined with Lemma 3, the following chain of inequalities
571 hold with the same probability

$$\left\| \widehat{\Sigma}_t^w - \Sigma(t/T) \right\|_{\infty/\infty} \leq \left\| \widehat{\Sigma}_t^w - \mathbb{E}[\Sigma_t^w] \right\|_{\infty/\infty} + \left\| \mathbb{E}[\Sigma_t^w] - \Sigma(t/T) \right\|_{\infty/\infty}$$
$$\leq \sqrt{\frac{\tau \log d}{cTh}} + C\left( h + \frac{1}{T^2 h^5} \right) \tag{30}$$

572 Replacing $h = T^{-1/3}$ in the above inequality gives rise to

$$\left\| \widehat{\Sigma}_t^w - \Sigma(t/T) \right\|_{\infty/\infty} \lesssim \frac{\sqrt{\tau \log d}}{T^{1/3}} \tag{31}$$

573 which completes the proof. $\qquad\square$

574 **Lemma 6.** *Assume that $h = T^{-1/3}$. Then, the following inequality holds for any $t$ and $\tau > 2$*

$$\left\| ST_\nu(\widehat{\Sigma}_t^w) - \Sigma(t/T) \right\|_\infty \lesssim \nu^{1-q} s(q,d) + \nu^{-q} s(q,d) \frac{\sqrt{\tau \log d}}{T^{1/3}} \tag{32}$$

575 *with probability of at least $1 - d^{-\tau+2}$.*

576 *Proof.* The proof is implied by Lemma 1 of [47] and Lemma 5. $\qquad\square$

577 *Proof of Corollary 4.* We only provide a sketch of the proof, due to to its similarity to the proof of
578 Corollary 3. One can write

$$\left\| \Theta(t/T) - [\mathrm{ST}_{\nu_t}(\widehat{\Sigma}_t^w)]^{-1} \right\|_{\infty/\infty} \leq \left\| [\mathrm{ST}_{\nu_t}(\widehat{\Sigma}_t^w)]^{-1} \right\|_\infty \|\Theta(t/T)\|_\infty \left\| \mathrm{ST}_{\nu_t}(\widehat{\Sigma}_t^w) - \Sigma(t/T) \right\|_{\infty/\infty}$$
$$\tag{33}$$

579 Due to Assumption 2, we have $\|\Theta(t/T)\|_\infty \leq \kappa_1$. Furthermore, similar to (18), one can write

$$\left\| \mathrm{ST}_{\nu_t}(\widehat{\Sigma}_t^w) - \Sigma(t/T) \right\|_{\infty/\infty} \leq \left\| \mathrm{ST}_{\nu_t}(\widehat{\Sigma}_t^w) - \widehat{\Sigma}_t^w \right\|_{\infty/\infty} + \left\| \widehat{\Sigma}_t^w - \Sigma(t/T) \right\|_{\infty/\infty}$$
$$\lesssim \frac{\sqrt{\tau \log d}}{T^{1/3}} \tag{34}$$

580 with probability of at least $1 - d^{-\tau+2}$, where the second inequality follows from Lemma 5 and
581 the choice of $\nu_t \asymp \frac{\sqrt{\tau \log d}}{T^{1/3}}$. Finally, Lemma 6 combined with an argument similar to the proof of
582 Corollary 3 leads to

$$\left\| [\mathrm{ST}_{\nu_t}(\widehat{\Sigma}_t^w)]^{-1} \right\|_\infty \leq \frac{2}{\kappa_2} \tag{35}$$

583 provided that

$$T \gtrsim s(q,d)^{\frac{3}{1-q}} (\tau \log d)^{3/2} \tag{36}$$

584 Combining these inequalities leads to the desired upper bound on (33). The rest of the proof is similar
585 to that of Corollary 3 and omitted for brevity. $\qquad\square$

## A.4 Proof of Proposition 1

Let $\delta_1 < \delta_2 < \ldots < \delta_m = T$ be the elements of the set $\Gamma$ from Algorithm 1, and define $\delta_0 = -1$. By construction, $\Delta^{\cap}_{\delta_{i-1}+1 \to \delta_i+1} = \emptyset$ for all $i = 1, \ldots, m-1$. It follows that for any $\theta$ satisfying bound constraints (7b) and $i = 1, \ldots, m-1$, we have that

$$\sum_{t=\delta_{i-1}+1}^{\delta_i} \mathbb{1}\{\theta_{t+1} - \theta_t \neq 0\} \geq 1.$$

Given any $j = 1, \ldots, T$, let $h$ be the maximum index such that $\delta_h < j$. Therefore, we find that for any feasible $\theta$,

$$f_{0 \to j}(\theta) = \sum_{t=0}^{j-1} \mathbb{1}\{\theta_{t+1} - \theta_t \neq 0\} \geq \sum_{t=0}^{\delta_h} \mathbb{1}\{\theta_{t+1} - \theta_t \neq 0\} = \sum_{i=1}^{h} \sum_{t=\delta_{i-1}+1}^{\delta_i} \mathbb{1}\{\theta_{t+1} - \theta_t \neq 0\} \geq h.$$

Since $f_{0 \to j}(\theta^{\texttt{Greedy}}) = h$ meets this lower bound, it follows that $\{\theta_t^{\texttt{Greedy}}\}_{t=0}^{j}$ is indeed an optimal solution to $\texttt{OPT}_{0 \to j}(1)$. Setting $j = T$ and $h = m - 1$, we find that $\theta^{\texttt{Greedy}}$ is optimal for $\texttt{OPT}_{0 \to T}(1)$. $\square$

## A.5 Proof of Theorem 5

Before proving this theorem, we need the following intermediate lemma:

**Lemma 7.** *Given any optimal solution $\widehat{\theta}$ to (7), exactly one of the following holds for any given zero-feasible sequence $\mathcal{Z}_{i \to j}$:*

*1. $\widehat{\theta}_i = \widehat{\theta}_{i+1} = \ldots = \widehat{\theta}_j = 0$*

*2. $\widehat{\theta}_\tau \neq 0$ for all $\tau = i, \ldots, j$.*

*Proof.* Let $\theta$ be any feasible solution to (3) that does not satisfy the conditions of Proposition 7, i.e., there exists $\tau = i, \ldots, j-1$ such that either $\theta_\tau = 0$ and $\theta_{\tau+1} \neq 0$, or $\theta_\tau \neq 0$ and $\theta_{\tau+1} = 0$. We now show how to construct a solution $\hat{\theta}$ with improved objective value, i.e., $f_{0 \to T}(\hat{\theta}) < f_{0 \to T}(\theta)$.

Consider the case $\theta_\tau = 0$ and $\theta_{\tau+1} \neq 0$. Define $\hat{\theta}_{\tau+1} = 0$ and $\hat{\theta}_t = \theta_t$ for all other coordinates $t \neq \tau + 1$. Clearly, $\hat{\theta}$ satisfies all bound constraints (3). Moreover,

$$f_{0 \to T}(\hat{\theta}) = f_{0 \to T}(\theta) - \underbrace{(1 - \gamma)}_{\hat{\theta}_{\tau+1} = 0} - \underbrace{\gamma}_{\hat{\theta}_\tau = \hat{\theta}_{\tau+1}} + \underbrace{\gamma \mathbb{1}\{\hat{\theta}_{\tau+1} \neq \hat{\theta}_{\tau+2}\}}_{\text{this term is 0 if } \tau + 1 = T} \leq f_{0 \to T}(\theta) - (1-\gamma) < f_{0 \to T}(\theta).$$

The case $\theta_\tau \neq 0$ and $\theta_{\tau+1} = 0$ is handled analogously. $\square$

Since Lemma 7 holds for any zero-feasible sequence, it holds in particular for all maximal zero-feasible sequences. Based on this lemma, we are ready to present the proof of Theorem 5.

*Proof of Theorem 5.* Let $\widehat{\theta}$ be an optimal solution to (7). Due to the optimality of $\widehat{\theta}$, the conditions in Lemma 7 are satisfied for all maximal nonzero intervals. We first show that there exists a path in $\mathcal{G}$ with cost $f(\widehat{\theta})$, and then we show that this path is indeed a shortest path.

Let $V_0 = \{v_1, v_2, \ldots, v_m\} \subseteq \{1, \ldots, Z\}$ be the set of indexes of the maximal zero-feasible sequences where $\widehat{\theta}$ vanishes, i.e., $\widehat{\theta}_{\mathcal{Z}_{i_s \to j_s}} = 0$ for every $s \in V_0$. It is easy to verify that $f^*$ is the optimal cost for the following constrained optimization:

$$f_{0 \to T}(\widehat{\theta}) = \min_{\{\theta_t\}_{t=0}^{T}} (1 - \gamma)\left(T + 1 - \left|\bigcup_{h=1}^{m} \mathcal{Z}_{i_{v_h} \to j_{v_h}}\right|\right) + \gamma \sum_{t=1}^{T} \mathbb{1}\{\theta_t - \theta_{t-1} \neq 0\} \qquad (37a)$$

$$\text{subject to} \quad l_t \leq \theta_t \leq u_t = 0, \ldots, T \qquad (37b)$$

$$\theta_t = 0 \qquad t \in \bigcup_{h=1}^{m} \mathcal{Z}_{i_{v_h} \to j_{v_h}}. \qquad (37c)$$

610   The constant term in (37a) reduces to

$$(1 - \gamma) \left( T + 1 - \left| \bigcup_{h=1}^{m} \mathcal{Z}_{i_{v_h} \to j_{v_h}} \right| \right) = (1 - \gamma) \left( T + 1 - \sum_{h=1}^{m} (j_{v_h} - i_{v_h} + 1) \right) \tag{38}$$

$$= (1 - \gamma) \left( i_{v_1} + \sum_{h=2}^{m} \left( i_{v_h} - j_{v_{h-1}} - 1 \right) + (T - j_{v_m}) \right). \tag{39}$$

611   Let the feasible region of (37) be denoted as $\mathcal{X}$. The second term in (37a), under constraints (37c),
612   decomposes as

$$\min_{\{\theta_t\}_{t=0}^{T} \in \mathcal{X}} \left\{ \gamma \sum_{t=1}^{T} \mathbb{1}\{\theta_t \neq \theta_{t-1}\} \right\}$$

$$= \gamma \min_{\{\theta_t\}_{t=0}^{T} \in \mathcal{X}} \left\{ \sum_{t=1}^{i_{v_1}} \mathbb{1}\{\theta_t \neq \theta_{t-1}\} \right\} + \gamma \sum_{h=1}^{m-1} \min_{\{\theta_t\}_{t=0}^{T} \in \mathcal{X}} \left\{ \sum_{t=j_{v_h}+1}^{i_{v_{h+1}}} \mathbb{1}\{\theta_t \neq \theta_{t-1}\} \right\}$$

$$+ \gamma \min_{\{\theta_t\}_{t=0}^{T} \in \mathcal{X}} \left\{ \sum_{t=j_{v_m}+1}^{T} \mathbb{1}\{\theta_t \neq \theta_{t-1}\} \right\} \tag{40}$$

613   Note that each intermediate term in (40) simplifies as follows:

$$\min_{\{\theta_t\}_{t=0}^{T} \in \mathcal{X}} \left\{ \sum_{t=j_{v_h}+1}^{i_{v_{h+1}}} \mathbb{1}\{\theta_t \neq \theta_{t-1}\} \right\} = \underbrace{\min_{\{\theta_t\}_{t=0}^{T} \in \mathcal{X}} \left\{ \sum_{t=j_{v_h}+2}^{i_{v_{h+1}}-1} \mathbb{1}\{\theta_t \neq \theta_{t-1}\} \right\}}_{=f^{\texttt{Greedy}}_{j_{v_h}+1 \to i_{v_{h+1}}-1}}$$

$$+ \underbrace{\mathbb{1}\{\theta_{j_{v_h}+1} \neq \theta_{j_{v_h}}\}}_{=1} + \underbrace{\mathbb{1}\{\theta_{i_{v_{h+1}}} \neq \theta_{i_{v_{h+1}}-1}\}}_{=1}. \tag{41}$$

614   Similarly, we find that the first and last term in (40) reduces to

$$\min_{\{\theta_t\}_{t=0}^{T} \in \mathcal{X}} \left\{ \sum_{t=1}^{i_{v_1}} \mathbb{1}\{\theta_t \neq \theta_{t-1}\} \right\} = \min_{\{\theta_t\}_{t=0}^{T} \in \mathcal{X}} \left\{ \sum_{t=1}^{i_{v_1}-1} \mathbb{1}\{\theta_t \neq \theta_{t-1}\} \right\} + \mathbb{1}\{i_{v_1} \geq 1\} = f^{\texttt{Greedy}}_{0 \to i_{v_1}-1} + \mathbb{1}\{i_{v_1} \neq 0\} \tag{42}$$

$$\min_{\{\theta_t\}_{t=0}^{T} \in \mathcal{X}} \left\{ \sum_{t=j_{v_m}+1}^{T} \mathbb{1}\{\theta_t \neq \theta_{t-1}\} \right\} = \min_{\{\theta_t\}_{t=0}^{T} \in \mathcal{X}} \left\{ \sum_{t=j_{v_m}+2}^{T} \mathbb{1}\{\theta_t \neq \theta_{t-1}\} \right\} + \mathbb{1}\{j_{v_m} + 1 \leq T\}$$

$$= f^{\texttt{Greedy}}_{j_{v_m}+1 \to T} + \mathbb{1}\{j_{v_m} < T\}. \tag{43}$$

615

616   Combining (43), (42) with (40) and (39), we find that $f_{0 \to T}(\widehat{\theta})$ is precisely the length of the path
617   $(0, v_1, \ldots, v_m, Z + 1)$ in the constructed graph $\mathcal{G}$ with weights defined as (8).

618   Now suppose that there exists a path $(0, \bar{v}_1, \bar{v}_2, \ldots, \bar{v}_p, Z + 1)$ with length $\bar{d} < f_{0 \to T}(\widehat{\theta})$. Consider
619   a solution $\bar{\theta}$ such that: (i) $\bar{\theta}$ is zero at zero-feasible sequences given by $\bar{v}_1, \bar{v}_2, \ldots, \bar{v}_p$, and (ii) $\bar{\theta}$ is
620   obtained from $\texttt{Greedy}(l, u, 0, i_{v_1} - 1)$, $\texttt{Greedy}(l, u, j_{v_p} + 1, T)$ and $\texttt{Greedy}(l, u, j_{v_h} + 1, i_{v_{h+1}} - 1)$,
621   otherwise. It is easy to verify that $\bar{\theta}$ is feasible and satisfies $f_{0 \to T}(\bar{\theta}) \leq \bar{d}$ (the inequality could be
622   strict if any solution reported by a call to the $\texttt{Greedy}$ routine has zero values), which contradicts the
623   optimality of $\widehat{\theta}$. Thus, we conclude that $f_{0 \to T}(\widehat{\theta})$ is indeed the length of the shortest $(0, Z + 1)$-path
624   in $\mathcal{G}$.                                                                                                        □

## A.6   Proof of Theorem 6

626   Algorithm 2 involves three main components: construct graph $\mathcal{G}$ (line 3), solve a shortest problem
627   on the constructed graph (line 4), and recover the optimal solution from the obtained shortest path.

628 Since $\mathcal{G}$ is acyclic, the shortest path problem can be solved in time linear in the number of arcs,
629 which is $\mathcal{O}(Z^2)$, via a simple labeling algorithm; see, e.g., Chapter 4.4. in [4]. Constructing graph
630 $\mathcal{G}$ requires computing the costs of all arcs. A naïve implementation, where Algorithm 1 is called
631 for every arc, would require $(O)(Z^2T)$ time and memory. However, from the second statement in
632 Proposition 1, we note that a single call to $\texttt{Greedy}(l, u, i, T)$ allows us to compute $f_{i \to j}^{\texttt{Greedy}}$ for all
633 $i \le j \le T$. Therefore, Algorithm 1 needs to be invoked only $\mathcal{O}(Z)$ times, and each call require $\mathcal{O}(T)$
634 leading to a total complexity of $\mathcal{O}(ZT)$. Moreover, given the shortest path, the optimal solution
635 can be constructed by concatenating the solutions obtained from the calls of $\texttt{Greedy}$. Finally, since
636 $Z \le T + 2$, we find that the overall complexity is dominated by that of constructing the graph. This
637 completes the proof. $\qquad\qquad\square$

## B   More on Numerical Experiments

639 In this section, we provide more information about the performance of the proposed estimator in
640 different case studies. In the first case study, our goal is to compare the statistical performance of our
641 proposed method with two other state-of-the-art methods, namely time-varying Graphical Lasso [17],
642 and a modified version of the elementary $\ell_1$ estimator [44, 47]. We will show that the proposed
643 estimator outperforms the other two estimators, in terms of both sparsity recovery and estimation
644 error. In the second case study, we showcase the statistical and computational performance of the
645 proposed method on massive-scale datasets. In particular, we will show that our proposed estimation
646 method can solve instances of the problem with more than 500 million variables in less than one hour,
647 with almost perfect sparsity recovery. Moreover, we demonstrate the improvements in the runtime
648 of our algorithm with parallelization. Finally, we conduct a case study on the correlation network
649 inference in stock markets. In particular, we show that the inferred time-varying graphical model can
650 correctly identify the stock market spikes based on the historical data.

651 All simulations are run on a desktop computer with an Intel Core i9 3.50 GHz CPU and 128GB RAM.
652 The reported results are for an implementation in MATLAB R2020b.

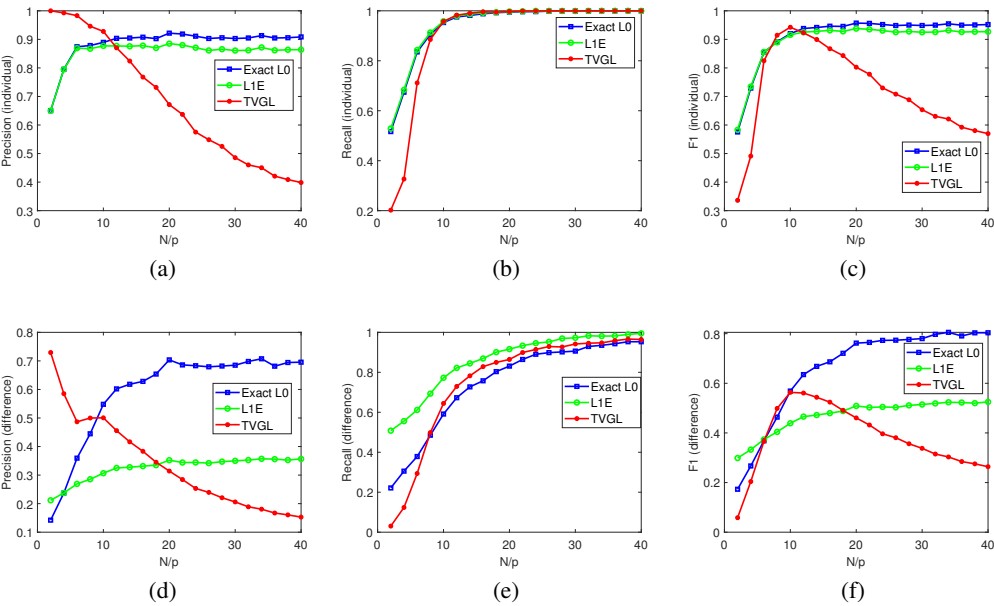

Figure 5: `Precision`, `Recall`, and `F1-score` for the estimated precision matrices and their differences using the proposed method (denoted as `Exact L`$_0$), `L1E`, and `TVGL` (averaged over 10 independent trials).

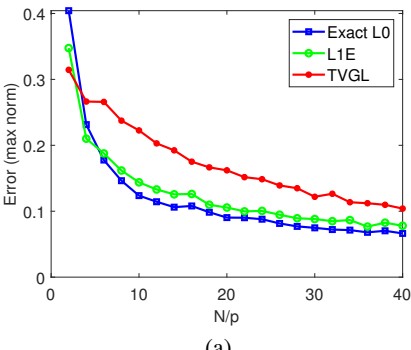 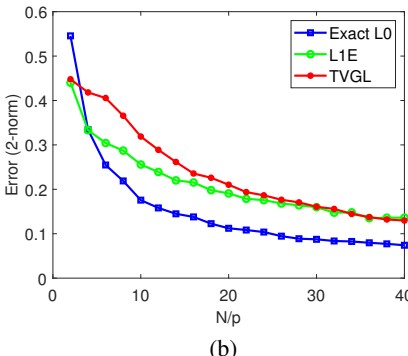

|(a)|(b)|

Figure 6: The normalized $\ell_\infty$-norm and induced 2-norm of the estimation error for the estimated precision matrices and their differences using the proposed method, L1E, and TVGL (averaged over 10 independent trials).

### B.1 Case Study on Small Datasets

In this case study, we evaluate the statistical performance of the proposed estimator, compared to two other methods, namely time-varying Graphical Lasso (TVGL) [17, 12], and a modified version of the elementary $\ell_1$ estimator (L1E) introduced in [44, 47]. As mentioned in the introduction, TVGL is a well-known regularized MLE approach for estimating the sparsely-changing GMRFs. On the other hand, different variants of L1E have been used to estimate static MRFs [47], and differential networks with sparsity imposed only on the parameter differences [44]. Consider an $\ell_1$ relaxation of the proposed estimator (3), where the $\ell_0$ penalties in the objective function are replaced with $\ell_1$ penalties. The resulted estimator reduces to that of [47] for $T = 0$, and [44] for $T = 1$ and $\gamma = 1$.

We consider randomly generated instances of sparsely-changing GMRFs, where the true inverse covariance matrix is constructed as follows: at time $t = 0$, we set $\Theta_0 = I_{d \times d} + \sum_{(i,j) \in \mathcal{S}} A^{(i,j)}$, where $d = 50$ and $A^{(i,j)}$ is a sparse positive semidefinite matrix with exactly two nonzero off-diagonal elements. In particular, we randomly select 100 edges in the graph (corresponding to 200 off-diagonal entries in $\Theta_0$) and collect their indices in $\mathcal{S}$. For every $(i, j) \in \mathcal{S}$, we set $A_{ij}^{(i,j)} = A_{ji}^{(i,j)} = -0.4$ and $A_{ii}^{(i,j)} = A_{jj}^{(i,j)} = 0.4$. Clearly, $A^{(i,j)} \succeq 0$, and hence, $\Theta_0 \succ 0$. Moreover, at every time $t = 1, \ldots, 9$, exactly 20 nonzero off-diagonal entries are added to $\Theta_t$ according to the aforementioned rules, and 20 nonzero nonzero off-diagonal entries are deleted by reversing the above procedure. Our goal is to estimate the true sparsely-changing precision matrices $\{\Theta_t\}_{t=0}^9$ based on a varying number of samples $N_t$. We evaluate the accuracy of the different methods in terms of Recall, Precision, and F1-score values, defined as

$$\texttt{Recall} = \frac{\texttt{TP}}{\texttt{TP} + \texttt{FP}}, \quad \texttt{Precision} = \frac{\texttt{TP}}{\texttt{TP} + \texttt{FN}}, \quad \texttt{F1-score} = \frac{2 \times \texttt{Recall} \times \texttt{Precision}}{\texttt{Recall} + \texttt{Precision}},$$
(44)

where TP, FP, and FN respectively denote the number of true positives, false positives, and false negatives in the estimated sequence of precision matrices. In all of our experiments, we set $\gamma = 0.7$. Moreover, according to Theorem 3, the parameters $\nu_t$ and $\lambda$ are chosen as $C_1 \sqrt{\frac{\log d}{T}}$ and $C_2 \sqrt{\frac{\log d}{T}}$, respectively, where the constants $C_1$ and $C_2$ are inferred directly from the data samples via Bayesian Inference Criterion (BIC) [32, 14]. Similarly, we set the regularization coefficients $\gamma_1 = C_3 \sqrt{\frac{\log d}{N_t}}$ and $\gamma_2 = C_4 \sqrt{\frac{\log d}{N_t}}$ for TVGL (2), where the constants $C_3$ and $C_4$ are selected via BIC.

Figure 5 illustrates the accuracy of the estimated precision matrices for different number of samples. It can be seen that the proposed estimator outperforms L1E and TVGL in terms of Precision value, but has a slightly worse Recall value. In particular, both L1E and TVGL tend to *overestimate* the number of nonzero elements in the precision matrices. This overestimation naturally reduces the number of false negatives (leading to better Precision values), while significantly increasing the

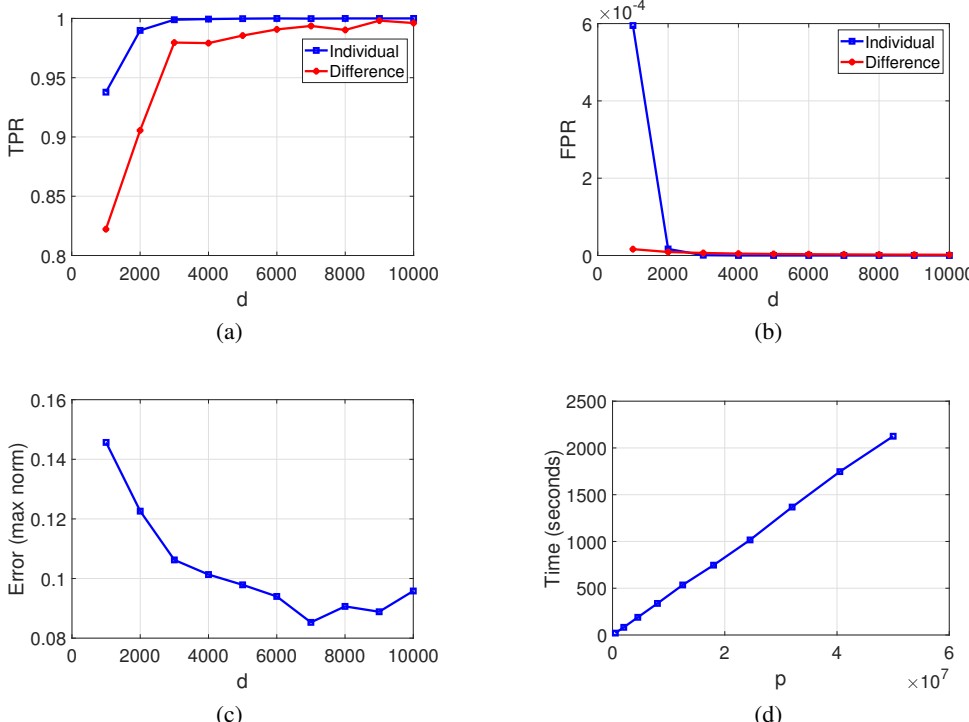

(a)

(b)

(c)

(d)

Figure 7: TPR, FPR, $\ell_1$-norm estimation error, and the runtime of the proposed method for fixed $T$ and different values of $d$. The number of samples $N_t$ is set to $d/2$ for every $t$. The runtime is shown with respect to $p = d(d+1)/2$.

number of false positives (leading to worse Recall values). Moreover, F1-score shows the overall performance of the estimates in terms of the sparsity recovery. It can be seen that the proposed estimator outperforms the other two methods. In particular, both L1E and TVGL perform poorly on the sparsity recovery of the parameter differences. Finally, Figure 6 depicts the normalized $\ell_\infty$-norm and induced 2-norm estimation errors. It can be seen that TVGL incurs a relatively large $\ell_\infty$-norm error due to the shrinking effect of its regularization.

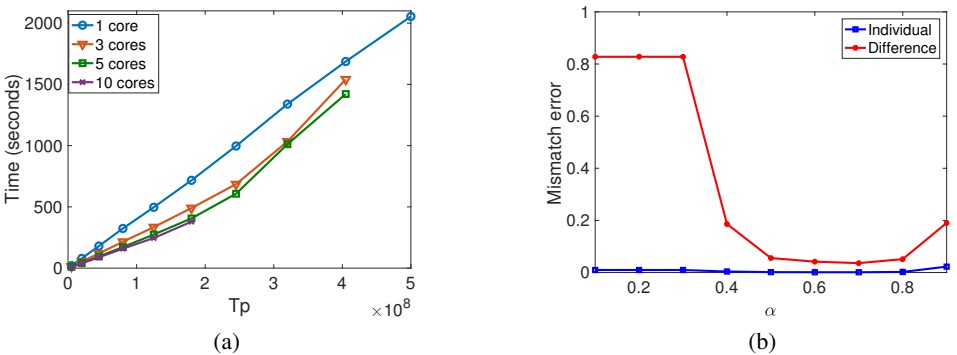

(a)

(b)

Figure 8: (a) The runtime of the parallelized algorithm with respect to the number of variables $Tp$, for different number of cores. (b) The normalized mismatch error with respect to the regularization coefficient $\gamma$, for the choices of parameters $d = 4000$, $T = 10$, and $N_t = 2000$ for every $t$.

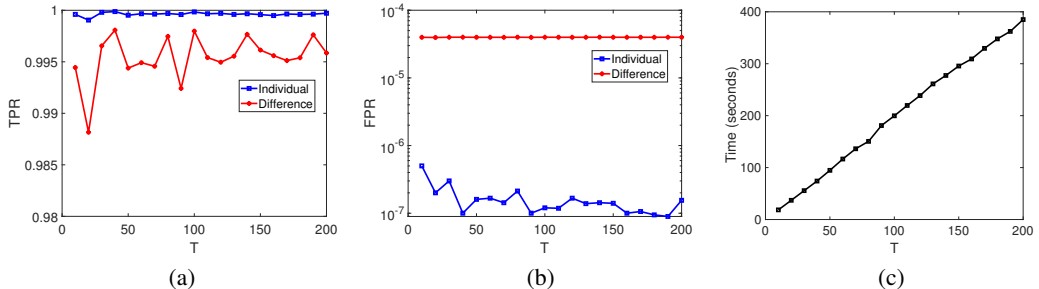

Figure 9: TPR, FPR, and the runtime of the proposed method for fixed $d$ and different values of $T$. The number of samples $N_t$ is set to $2d$ for every $t$.

## B.2 Case Study on Large Datasets

In this case study, we analyze the performance of the proposed estimator on large datasets, with different values of $d$ and $T$. In particular, we will analyze the runtime of the proposed algorithm and its statistical performance in high dimensional settings, where $N_t < d$ for every $t = 1, 2, \ldots, T$. Moreover, we will report the improvements in the runtime with parallelization, and analyze the robustness of the estimator for different choices of the regularization coefficient $\gamma$.

Consider the class of synthetically generated sparsely-changing GMRFs with random precision matrices, as explained in Subsection B.1. In the first experiment, we fix $T = 10$ and change the values of $d$. The number of nonzero elements in the individual precision matrices and their differences are set to $3d$ and $0.04d$, respectively. We evaluate the performance of the proposed method in the high dimensional settings, where $N_t = d/2$ for every $t = 0, \ldots, T$. The parameters $\lambda_t$ and $\nu_t$ are fine-tuned similar to the previous case study and $\gamma = 0.7$ in all instances. Moreover, define TPR and FPR for the individual parameters and their differences as the TP and FP values, normalized by the total number of nonzero and zero elements in the true precision matrices and their differences, respectively. Clearly, both TPR and FPR are between 0 and 1, with TPR = 1 and FPR = 0 corresponding to the perfect recovery of the sparsity patterns. Figure 7 depicts TPR, FPR, and the $\ell_1$-norm error of the estimated parameters, as well as the runtime of our algorithm for different values of $d$. It can be seen that both TPR and FPR values improve with the dimension for the estimated parameters and their differences. Moreover, the runtime of our algorithm scales almost linearly with $p = d(d+1)/2$, which is in line with the result of Theorem 2. Using our algorithm, we reliably infer instances of sparsely-changing GMRFs with more than 500 million variables in less than one hour.

As mentioned before, our proposed optimization framework is amenable to parallelization due to its elementwise decomposable nature. Figure 8a illustrates the runtime of our parallelized algorithm with respect to the total number of variables (fixed $T$ and varying $p$), for different number of cores. Using 5 cores, the runtime of our algorithm is improved by $40\%$ on average. On the other hand, using 10 cores deteriorates the performance due to the shared memory limitations. Finally, we evaluate the accuracy of the estimated parameters for different choices of the regularization coefficient $\gamma$. In particular, we fix $d = 4000$, $T = 10$, and $N_t = 2000$ for every $t$, and depict the normalized mismatch error in the sparsity pattern of the estimated parameters and their differences for $\gamma \in \{0.1, 0.2, \ldots, 1\}$. Based on this figure, it can be concluded that overall performance of the proposed method is not too sensitive to specific choice of the regularization parameter $\gamma$. In particular, it can be seen that the normalized mismatch error remains approximately the same for $\gamma \in [0.5, 0.8]$.

In the next experiment, we set $d = 1000$ and $N_t = 2d$, and evaluate the performance of the proposed method for different values of $T \in \{10, 20, 30, \ldots, 200\}$. Figure 9a shows TPR for the estimated precision matrices and their differences. It can be seen that TPR for the estimated precision matrices is close to 1 for all values of $T$. Moreover, the TPR for the differences of the estimated precision matrices is at least $0.966$. On the other hand, Figure 9b shows that the FPR for the estimated precision matrices is close to zero. Finally, Figure 9c shows that the runtime of the proposed algorithm scales almost linearly with $T$. Together with Figure 7d, this implies that the empirical complexity of the algorithm is linear in both $p$ and $T$.

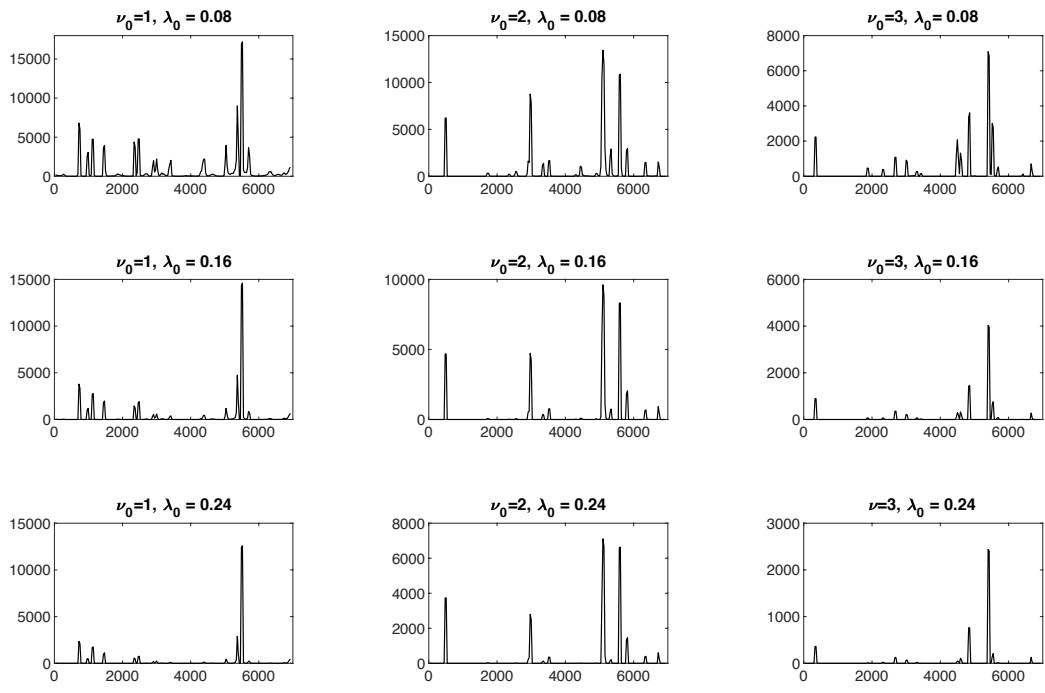

Figure 10: The number of changes in the estimated stock correlation network, for different choices of $\nu_0$ and $\lambda_0$. The $x$-axis represent the day indexes.

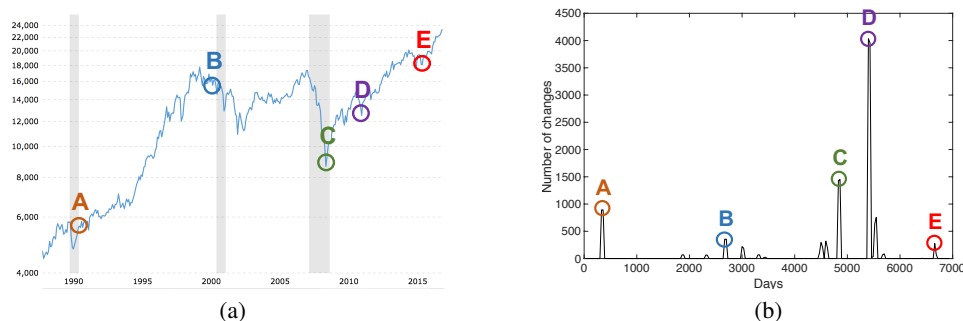

(a)                                                        (b)

Figure 11: (a) NASDAQ historical chart from 1988 to 2017 [2]. (b) The number of changes in the estimated correlation network for $\nu_0 = 3$ and $\lambda_0 = 0.16$.

## B.3 Case Study on Stock Market

Finally, we illustrate the performance of our algorithm for the inference of stock correlation network. We consider the daily stock prices for 214 securities from $1990/01/04$ to $2017/08/10$, with the total number of 6990 days ($d = 214$ and $T = 6990$). Due to the continuously changing nature of the stock correlation network, we will use the kernel averaging approach that was introduced in Subsection 4.1 to estimate the underlying time-varying network. In particular, we consider a Gaussian kernel with bandwidth $h = 0.3T^{-1/3}$ to obtain the sequence of weighted sample covariance matrices. Using the constructed sample covariance matrices, we estimate the sparsely-changing precision matrix $\Theta(t/T)$ at discrete times $t \in \{30, 60, 90, \ldots, 6990\}$. Moreover, we set $\gamma = 0.9$, $\lambda_t = \lambda_0\sqrt{\frac{\log(d)}{Th}}$, and $\nu_t = \nu_0\sqrt{\frac{\log(d)}{Th}}$, for some constants $\lambda_0$ and $\nu_0$ to be defined later. Note that these choices of the parameters are consistent with the assumptions of Theorem 4.

Figure 10 shows the number of changes in the sparsity pattern of the estimated correlation network, for different choices of the parameters $\nu_0$ and $\lambda_0$. A drastic change in the correlation network signals a *spike* in the stock market, which may reflect the market's response to unexpected global events. It can be seen that, for small values of $\nu_0$ and $\lambda_0$, the estimated network can detect both small and large spikes. As the values of $\nu_0$ and $\lambda_0$ increase, the small spikes gradually dimish, and the estimated network only "picks up" major changes in the network. Nonetheless, there is a recurring pattern of spikes in these plots that is almost insensitive to different values of $\nu_0$ and $\lambda_0$. A closer look at this recurring pattern sheds light on the behavior of the market. Figure 11 shows the number of changes in the estimated network, for the choices of $\nu_0 = 3$ and $\lambda_0 = 0.16$, together with the historical chart of National Association of Securities Dealers Automated Quotations (NASDAQ) [1]. It can be seen that the major spikes in the estimated network can be attributed to the historical stock market *crashes*. For instance, the spikes A, B, and C respectively correspond to the "early 1990s recession", "dot-com bubble", and "global financial crisis"; see [5] for more details. Interestingly, the estimated network can also detect other historical (but less severe) downturns in 2011 (point D) and 2016 (point E).