# OpenReview forum: "Scalable Inference of Sparsely-changing Gaussian Markov Random Fields "
_NeurIPS.cc/2021/Conference — NeurIPS 2021 Poster_

### Official Review · Reviewer_CuaR · 2021-07-16

**Rating:** 7
**Confidence:** 4

**Summary:**

This work tackles the sparsely changing GMRT problems. By mainly comparing with TVGL, the method claims to have stronger theoretical guarantees and computational efficiency.

**Limitations And Societal Impact:**

It would be interesting to apply this method to some COVID dataset for a bigger societal impact.

**Main Review:**

Originality: The problem setup is mainly covered in TVGL paper, but the method is novel enough in a sense of proposing new optimization problem, optimization algorithm, and statistical guarantees.

Quality/Significance: It provided solid and thorough theoretical guarantees and proofs. However, the experiments are rather weak. It only compared with TVGL through one simple synthetic experiments (that could have been particularly adversarial) whereas TVGL did extensive synthetic and real data experiments with reporting various scores.

Clarity: The writing can be improved in general. To name a few, for example, indicate that the the proof of theorem 2 will be provided in section 5. Some notations like approx backward mapping \tilde F, \Delta^cap_{t->s} is overly complicated, making the readers difficult to follow.

Additional questions to be clarified:
1. (3c) should be symmetric. How do you force it in optimization?
2. In line 88, TVGL does not use interior point method, only requiring O(Td^3) per iteration through ADMM that can be parallelized.
3. In line 124, the matrix inversion seems to be needed for each time t. and thus need Td^3 in total? It this is true, how do you compare the computational cost with TVGL?
4. In line 190, there must be some sufficient condition on \nu to define its inverse properly, e.g. positive, etc.



**Time Spent Reviewing:**

2

---

> ### Author Response · Authors · 2021-08-09
> **Response to Reviewer CuaR**
>
> **C** "However, the experiments are rather weak. It only compared with TVGL through one simple synthetic experiments (that could have been particularly adversarial) whereas TVGL did extensive synthetic and real data experiments with reporting various scores."
>
> **R:** We greatly appreciate the reviewer's insightful comment. We kindly refer the reviewer to our response to Reviewer 5AUA, where we have provided a detailed comparison between different approaches with respect to different scores. Moreover, we will also compare the performance of TVGL and our method on other realistic datasets.
>
> We would also like to point out that our comparison is not based on one specific instance of the problem. All of our simulations are averaged over 10 independent trials (leading to more than 200 experiments in total). Moreover, we mainly focused on synthetically generated datasets with known ground truth to assess the statistical performance of different estimators with respect to various metrics.
>
> **C:** "The writing can be improved in general. To name a few, for example, indicate that the the proof of theorem 2 will be provided in section 5. Some notations like approx backward mapping $\tilde F$, $\Delta^{\cap}_{t\to s}$ is overly complicated, making the readers difficult to follow."
>
> **R:** Thank you for this constructive comment. We will improve the writing and the used notations to further enhance the readability of the paper.
>
> **C:** "(3c) should be symmetric. How do you force it in optimization?"
>
> **R:** We only solve the problem for the upper triangular part of the matrix variables, and then complete the matrix by preserving their symmetric nature.
>
> **C** "In line 88, TVGL does not use interior point method, only requiring $O(Td^3)$ per iteration through ADMM that can be parallelized."
>
> **R:** The reviewer is indeed correct. However, note that ADMM suffers from notoriously slow convergence rate of $O(1/\epsilon)$, thereby increasing the overall complexity to $O(Td^3/\epsilon)$. An important implication of this slow convergence rate is that the algorithm may require exponential number of iterations to improve the accuracy of the solution by a single digit. This is in stark contrast with our method, which enjoys a fully polynomial algorithm. For instance, taking a rather loose accuracy $\epsilon = 1/T$ would increase the theoretical complexity of ADMM to $O(T^2 d^3)$, which is strictly worse than our proposed method.
>
> From a practical point of view, solutions obtained from ADMM-based approaches suffer from low accuracy. In fact, even the tailored ADMM algorithm proposed for TVGL uses a rather loose accuracy of $\epsilon = 10^{-3}$ (this is not explicitly mentioned in the paper \[17\], and instead, is encoded in their algorithm; see <https://github.com/davidhallac/TVGL> for more details).
>
> Finally, note that in the parallelized ADMM, each thread needs to solve a large conic optimization problem over a $d\times d$ matrix. The complexity of solving this single sub-problem varies from $O(d^3/\epsilon)$ (via first-order methods) to $O(d^6\log(1/\epsilon))$ (using interior-point methods). In contrast, our aproach requires solving a small problem with $O(T)$ variables on each thread, each solvable in $O(T^2)$ time and memory.
>
> **C:** "In line 124, the matrix inversion seems to be needed for each time t. and thus need $Td^3$ in total? It this is true, how do you compare the computational cost with TVGL?"
>
> **R:** We sincerely thank the reviewer for pointing out this typo. Indeed, our algorithm needs the approximate backward mapping at every time $t = 0,\dots, T$, and each approximate backward mapping requires a single matrix inversion. Therefore, the worst-case complexity of obtaining all backward mappings is $O(T d^3)$. Given these backward mappings, our algorithm can solve the problem to optimality in $O((dT)^2)$ (see Theorem 2), leading to an end-to-end complexity of $O((dT)^2 +Td^3)$. Moreover, one cannot avoid the complexity of $O(Td^3)$, even if the true sample covariances are available (since we still need to invert them to obtain the true precision matrices). In light of this, the complexity of our algorithm coincides with this lower bound, provided that $T = O(d)$, which holds in many practical problems, such as brain networks, financial markets, and genomics.
>
> In contrast, ADMM for TVGL requires an eigendecomposition at every time $t = 0,1,\dots, T$, and for every iteration $k = 1,2,\dots,\lceil 1/\epsilon\rceil$, leading to a total of $O(T/\epsilon)$ number of eigendecompositions (see \[17\] for more details on implementation). Each eigendecomposition has the complexity of $O(d^3)$ (same as matrix inversion), bringing the total complexity to $O(Td^3/\epsilon)$. In contrast, our proposed method requires inverting exactly $T$ sparse matrices (which is significantly smaller than $O(T/\epsilon)$), which can be done in parallel and in an offline fashion before running our algorithm.
>
> **C** "In line 190, there must be some sufficient condition on $\nu$ to define its inverse properly, e.g. positive, etc."
>
> **R:** The reviewer is indeed correct that the parameter $\nu_t$ must be greater than a threshold to ensure the invertibility of the thresholded sample covariance matrix. Here, we show that this is indeed the case for our choice of $\nu_t$. To see this, note that a modified version of Lemma 2 in the appendix, together with the choice of $\nu_t$ ensures that $\|ST_{\nu}(\widehat\Sigma)-\Sigma\|\lesssim \sqrt{\log d/N_t}$, where $ST_{\nu}(\widehat\Sigma)$ is the thresholded sample covaraince and $\Sigma$ is the true covariance. By Weyl's inequality, we have $|\hat\lambda_{\min}-\lambda_{\min}|\leq \|ST_{\nu}(\widehat\Sigma)-\Sigma\|\lesssim \sqrt{\log d/N_t}$, where $\hat\lambda_{\min}$ and $\lambda_{\min}$ are the minimum eigenvalues of $ST_{\nu}(\widehat\Sigma)$ and $\Sigma$, respectively. Therefore, we have $\hat\lambda_{\min}>0$, provided that $N_t\gtrsim \log d/\lambda_{\min}^2\asymp \log d$. This implies that the thresholded sample covariance is positive definite, and hance, invertible. We will clarify this point in the revised paper.

---

### Official Review · Reviewer_GycU · 2021-07-16

**Rating:** 6
**Confidence:** 3

**Summary:**

This paper proposes a scalable algorithm to learn time varying gaussian graph models, and show that it has good convergence property. It also conducts a large scale experiment to demostrate the efficiency of the algorithm.

**Limitations And Societal Impact:**

The notations of matrix norms should be defined more clearly. I can only guess the double infinity norm is the entrywise norm? I have not seen this notation before.

In the experiments, the authors should provide more comparisons to previous methods. Is the time-varying Graphical Lasso applicable to the large scale problem? If so, what is the performance of it compared to the proposed algorithm?

**Main Review:**

The theoretical findings of this paper is very good. The algorithm is both simple and efficient. And it will be useful for a wide range of problems.

The structure of the paper is somewhat counterintuitive. The definitions of $F^*$ and the main idea of the algorithm should be moved to the beginning of the paper. Otherwise some statements in Section 2 are quite confusing.

**Time Spent Reviewing:**

4

---

> ### Author Response · Authors · 2021-08-09
> **Response to Reviewer GycU**
>
> **C:** "The structure of the paper is somewhat counterintuitive. The definitions of $F^*$ and the main idea of the algorithm should be moved to the beginning of the paper. Otherwise some statements in Section 2 are quite confusing."
>
> **R:** We sincerely apologize for this confusion. We will restructure the paper according to the reviewer's comment, and ensure that the main idea of the algorithm, as well as the related definitions are presented at the earlier parts of the paper.
>
> **C:** "The notations of matrix norms should be defined more clearly. I can only guess the double infinity norm is the entrywise norm? I have not seen this notation before."
>
> **R:** We have already introduced different notations and symbols used throughout the paper (including different definitions of matrix norms) in Notation section. The reviewer is kindly referred to the second page of the paper (lines 40-48).
>
> **C:** "In the experiments, the authors should provide more comparisons to previous methods. Is the time-varying Graphical Lasso applicable to the large scale problem? If so, what is the performance of it compared to the proposed algorithm?"
>
> **R:** We greatly appreciate the reviewer's insightful comment. We would like to point out that none of the alternative approaches (including time-varying Graphical Lasso) converge in the scales considered in our paper. Moreover, these alternative approaches suffer from inferior statistical performance. We kindly refer the reviewer to our response to Reviewer 5AUA, where we have addressed this comment in detail.

---

### Official Review · Reviewer_zSRw · 2021-07-16

**Rating:** 8
**Confidence:** 4

**Summary:**

This submission considers the problem of estimating the time-varying precision matrices corresponding to a Gaussian graphical model. The authors propose an \ell_0-regularized optimization problem (that runs in polynomial time), and establish theoretical guarantees as well as its performance on simulated and real datasets. The paper is quite well-written and seems like an important contribution to this active research area.

**Limitations And Societal Impact:**

This is primarily a theoretical paper, and I do not foresee any potential negative societal impacts.

**Main Review:**

There has been considerable interest in establishing efficient algorithms and theoretical guarantees for recovering the underlying structure (i.e., edges) of Markov random fields (MRFs). A great deal of this work has focused on the Gaussian case, and relied on \ell_1-regularized maximum likelihood estimators, often referred to as Graphical Lasso. Most of this work makes so-called "incoherence" assumptions on the underlying precision matrix (and this work seems to rely on similar assumptions) to then give sample complexity bounds.

In many settings, it may be of more interest to understand how the edges of the MRF change over time. The naive approach to this problem is to re-estimate the set of edges, and compare the differences. However, the sample complexity will be dominated by the total number of edges in each instance, even if the number of changes is small. This submission focused on this problem and demonstrates that an \ell_0-regularized maximum likelihood estimator can be solved in polynomial time to find the changes, with a sample complexity that depends primarily on the number of changes. Thus, the paper makes a novel conceptual contribution and uses to establish new theoretical guarantees on a well-motivated problem. To the best of my knowledge, this is the first paper to characterize the sample complexity of time-varying precision matrices via an efficient algorithm.

The paper also demonstrates the performance of the proposed \ell_0-regularized estimator on simulated and real data. In particular, I appreciated Figure 1, which clearly establishes that traditional \ell_1-regularized methods fail to capture the sparse, time-varying changes, combined with Figure 2 that shows the success of the proposed method.

Overall, the paper is well-written and well-placed in the context of the literature. My main questions with the paper are as follows:

1. How do the precision matrix assumptions in this work compare to those of related work? Is there a natural family of models where both sets of assumptions apply? How efficiently can these assumptions be checked for a given model?

2. I didn't quite understand the point of Figure 3. Wouldn't it be helpful to plot the \ell_1-performance to make a comparison? If this is not feasible (due to runtime issue), then a comment to that effect would suffice.



**Time Spent Reviewing:**

2

---

> ### Author Response · Authors · 2021-08-09
> **Response to Reviewer zSRw**
>
> **C:** "How do the precision matrix assumptions in this work compare to those of related work? Is there a natural family of models where both sets of assumptions apply?"
>
> **R:** We are thankful for the reviewer's insightful questions. Our main results on the statistical performance of the proposed method relies on the notion of weak sparsity. In particular, rather than imposing a condition on the weak sparsity, we use this notion as a *parameter* to characterize the sample complexity of our method. In other words, our theoretical result still holds for large values of $s(q,d)$ (which captures the level of weak sparsity), provided that the number of samples scale accordingly. In contrast, most existing guarantees on graphical models work under the so-called *mutual incoherence* condition on the true precision matrices, which may not hold even with infinitely many samples. Indeed, one of the main advantages of our proposed method is that it eschews the need for such conditions by resorting to a more tractable formulation.
>
> Moreover, in addition to the class of matrices that are mentioned in the paper, a more general class of precision matrices with weakly sparse structures are those with strictly diagonally dominant patterns. In particular, suppose that $\Theta$ is a sparse precision matrix with strictly diagonally dominant structure. In particular, define $\Delta_i = \Theta_{ii}-\sum_{j\not=i}|\Theta_{ij}|>0$. Then, according to \[R1\], we have $\sum_{j=1}^d|\Sigma_{kj}| = \sum_{j=1}^d|[\Theta^{-1}]_{ij}|\leq \frac{1}{\min_i\{\Delta_i\}}$, for every $k=1,\dots, d$. In other words, the covariance matrix is weakly sparse, provided that $\Delta_i$ does not scale down to zero with the dimension. The diagonally dominant structures naturally arise in the context of graphical model inference; see \[R2\] for various applications, including those with graph Laplacian structures.
>
> \[R1\] Varah, James M. \"A lower bound for the smallest singular value of a matrix.\" Linear Algebra and its applications 11.1 (1975): 3-5.
>
> \[R2\] Egilmez, H. E., Eduardo Pavez, and Antonio Ortega. \"Graph Learning from Data under Structural and Laplacian Constraints.\" IEEE Journal of Selected Topics in Signal Processing (2017).
>
> **C:** "How efficiently can these assumptions be checked for a given model?""
>
> **R:** We would like to point out that, while our algorithm does not directly depend on the weak sparsity parameter $s(q,d)$, it can be estimated from the available data using the proposed thresholded sample covariance matrix. Here, we briefly explain our approach to estimate $s(q,d)$. According to Lemmas 2 and 6 in the appendix, the thresholded sample covriance is close to its true counterpart, provided that $N_t\gtrsim \log d$ (or $T\gtrsim (\log d)^{1.5}$ with kernel averaging). Under such circumstances, one can estimate the weak sparsity parameter s(q,d) by directly computing it for the thresholded sample covaraince matrix. Denoting this surrogate quantity by $\widehat s(q,d)$, one can guarantee that $|s(q,d)| - \widehat s(q,d)|$ is small due to the closeness of the thresholded sample covaraince matrix to its true counterpart. We will make this argument precise in the revised paper.
>
>
>
>
> **C:** "I didn't quite understand the point of Figure 3. Wouldn't it be helpful to plot the L1-performance to make a comparison? If this is not feasible (due to runtime issue), then a comment to that effect would suffice."
>
> **R:** We greatly appreciate the reviewer's useful comment. We would like to point out that none of the alternative approaches (including the elementary $\ell_1$ estimator tackled using an off-the-shelf solver) converge in the scales considered in Figure 3. We kindly refer the reviewer to our response to Reviewer 5AUA, where we have discussed and compared the statistical and computational performance of different methods in detail.

---

### Official Review · Reviewer_jzT4 · 2021-07-17

**Rating:** 7
**Confidence:** 3

**Summary:**

In this paper, the authors consider the problem of learning Sparsely changing Gaussian Markov Random Fields. They pose the problem as a non-convex optimization problem that can be solved exactly. The authors provide statistical guarantees. Furthermore, they also provide a computationally efficient algorithm to recover the graphical model.

**Limitations And Societal Impact:**

The paper does not seem to have any potential negative societal impact.

**Main Review:**

Originality – The problem of Gaussian Graphical Models evolving with time is surely an interesting one and the authors provide a novel optimization problem formulation to solve this problem. The authors clearly delineate their work compared to existing literature.

Quality – The claims are well supported. The first concern I have is with the weak sparsity condition. While the authors do provide some insights into the examples when the weak sparsity is bounded, it seems to restrict the applicability to a small class of Gaussian graphical models. Another concern is the lack of discussion about choosing the parameters $\lambda$ and $\gamma$.

Clarity – The paper is well written and easy to follow.

Significance – The paper addresses an important problem and the practicality of the proposed solution could see it being used in practice. Future work could be aimed at demystifying and hopefully relaxing the weak sparsity condition.

Overall I think this is a great paper with a novel approach to solve an interesting problem and has a good mix of statistical guarantees as well as a computationally feasible algorithm.

-------------------------After Rebuttal-------------------------
Thanks for providing further insights into the weak sparsity condition as well as choosing $\lambda$ and $\gamma$, it really helped in bringing out the idea clearly.


**Time Spent Reviewing:**

12

---

> ### Author Response · Authors · 2021-08-09
> **Response to Reviewer jzT4**
>
> **C:** "The first concern I have is with the weak sparsity condition. While the authors do provide some insights into the examples when the weak sparsity is bounded, it seems to restrict the applicability to a small class of Gaussian graphical models."
>
> **R:** We truly appreciate the reviewer's insightful comment. We would like to point out that, rather than imposing a condition on the weak sparsity, we use this notion as a *parameter* to characterize the sample complexity of our method. In other words, our theoretical result still holds for large values of $s(q,d)$ (which captures the level of weak sparsity), provided that the number of samples scale accordingly. In contrast, most existing guarantees on graphical models work under the so-called *mutual incoherence* condition on the true precision matrices, which imposes a restrictive condition on the Hessian of the log-likelihood function, and may not hold even with infinitely many samples. Indeed, one of the main advantages of our proposed method is that it eschews the need for such conditions by resorting to a more tractable formulation.
>
> Moreover, in addition to the class of matrices that are mentioned in the paper, a more general class of precision matrices with weakly sparse structures are those with strictly diagonally dominant patterns. In particular, suppose that $\Theta$ is a sparse precision matrix with strictly diagonally dominant structure. In particular, define $\Delta_i = \Theta_{ii}-\sum_{j\not=i}|\Theta_{ij}|>0$. Then, according to \[R1\], we have $\sum_{j=1}^d|\Sigma_{kj}| = \sum_{j=1}^d|[\Theta^{-1}]_{ij}|\leq \frac{1}{\min_i\{\Delta_i\}}$, for every $k=1,\dots, d$. In other words, the covariance matrix is weakly sparse, provided that $\Delta_i$ does not scale down to zero with the dimension. The diagonally dominant structures naturally arise in the context of graphical model inference; see \[R2\] for various applications, including those with graph Laplacian structures. We will address the reviewer's comment in the revised paper by carefully delineating these points.
>
> \[R1\] Varah, James M. \"A lower bound for the smallest singular value of a matrix.\" Linear Algebra and its applications 11.1 (1975): 3-5.
>
> \[R2\] Egilmez, H. E., Eduardo Pavez, and Antonio Ortega. \"Graph Learning from Data under Structural and Laplacian Constraints.\" IEEE Journal of Selected Topics in Signal Processing (2017).
>
> **C:** "Another concern is the lack of discussion about choosing the parameters $\lambda$ and $\gamma$."
>
> **R:** We thank the reviewer for raising this point, and apologize for the lack of discussion on the choice of the parameters. In all of our simulations, the parameters $\lambda$ and $\gamma$ are chosen based on Bayesian Inference Criterion (BIC). In particular, according to Theorem 3, the parameters $\nu$ and $\lambda$ must be chosen as $C_1\sqrt{\frac{\log d}{N_t}}$ and $C_2\sqrt{\frac{\log d}{N_t}}$, respectively, where $C_1$ and $C_2$ are scalars that depend on the unknown parameters of the problem. Since we do not have access to these parameters a priori, we estimate them directly from the data via BIC. In particular, we choose the constants $C_1$ and $C_2$ that maximize a penalized variant of the empirical likelihood function. This insures that the chosen parameters "automatically select" the best model implied by the available data, while avoiding overfitting. For more details on the exact formulation of BIC, the reader is kindly referred to \[14\] and \[32\]. We will better describe our parameter selection method in the revised manuscript.

---

### Official Review · Reviewer_5AUA · 2021-07-17

**Rating:** 6
**Confidence:** 2

**Summary:**

  This paper study the problem of inferring time-varying Gaussian Markov random fields, where the underlying graphical model is both sparse and changes {sparsely} over time. A new class of constrained optimization problems was introduced for the inference of sparsely-changing Gaussian MRFs (GMRFs). The proposed optimization problem is formulated based on the exact regularization and can be solved in near-linear time and memory.



**Limitations And Societal Impact:**

Yes

**Main Review:**

The reviewer finds this work provides very interesting and meaningful theoretical and algorithmic results. The overall novelty and significance is very good and would bring significant impact to various real-world application.

However, the reviewer does find numerical tests compared with existing works missing, as only the performance evaluated is for the proposed algorithm, but no comparison with other existing frameworks is provided.

**Time Spent Reviewing:**

3

---

> ### Author Response · Authors · 2021-08-09
> **Response to Reviewer 5AUA**
>
> **C:** "However, the reviewer does find numerical tests compared with existing works missing, as only the performance evaluated is for the proposed algorithm, but no comparison with other existing frameworks is provided."
>
> **R:** We thank the reviewer for this insightful comment. We would like to point out that in the appendix, we compare our proposed method with two other state-of-the-art methods, namely time-varying Graphical Lasso (TVGL) \[17\] (which is one of the most commonly used approaches for the inference of time-varying MRFs) and elementary L1 estimator \[44, 47\] (which is a recently proposed alternative to TVGL); see Section B and B1 in the appendix for more details. In what follows, we summarize our simulations.
>
> *Statistical Performance:* Our proposed method outperforms both TVGL and elementary L1 estimator in terms of estimation error, as well as different metrics such as F1-score. Moreover, to better illustrate the performance of TVGL, we take a closer look at its behavior in a simple case study in Example 1 (see pages 2 and 4). In particular, we show that TVGL suffers from high mismatch error and excessive bias, caused by the shrinkage effect of L1 regularization.
>
> *Computational Time:* In addition to the superior statistical performance, our proposed method is more efficient than the existing off-the-shelf solvers for TVGL and elementary L1 estimator. In fact, we believe that this is the main advantage of our method compared to the existing approaches. The following table shows the runtime of our algorithm, compared to that of TVGL with three different solvers, namely MOSEK, SEDUMI, and SDPT3, as well as the runtime of the elementary L1 estimator. In our simulations, we set T = 10 and change d from 25 to 5000. All runtimes are reported in seconds, and averaged over 3 trials. The symbol "-" implies that the solver failed or did not converge within 3 hours.
>
> | Method (Solver)       | d=25  | d=50    | d=100   | d=150 | d=500  | d=5000 |
> |-----------------------|-------|---------|---------|-------|--------|--------|
> | TVGL (MOSEK)          | 5.82  | 63.59   | 1570.87 | -     | -      | -      |
> | TVGL (SEDUMI)         | 58.01 | 2831.15 | -       | -     | -      | -      |
> | TVGL (SDPT3)          | 90.74 | -       | -       | -     | -      | -      |
> | Elementary L1 (MOSEK) | 2.01  | 2.45    | 12.68   | 29.47 | 453.12 | -      |
> | Our method            | 0.12  | 0.21    | 0.46    | 0.72  | 5.68   | 501.87 |
>
> Based on the above table, our proposed method is on average 1255 times faster than the best solver for TVGL, and 36 times faster than elementary L1 estimator (among the instances that can be solved to optimality via both methods). Based on these simulations, we hope to have convinced the reviewer that our proposed inference method is a more scalable alternative for solving time-varying MRFs in massive scales. We will add this discussion to the revised version of the paper to better address the reviewer's comment.

---

### Author Response · Authors · 2021-08-09
**General response to reviewers**

We are grateful to the reviewers for their time and effort in reviewing our paper. We are happy to receive an overall positive response, and are grateful for their valuable comments and suggestions for improvement. We have addressed the reviewers' comments individually below

---

### Decision · Program_Chairs · 2021-09-27

**Decision:**

Accept (Poster)

**Comment:**

This paper considers the task of estimating a sparse Gaussian graphical model with a graph that changes sparsely over time. A nonconvex optimization formulation is shown to be efficiently solvable and to achieve good performance both empirically and theoretically. This is an important problem and the paper makes worthwhile progress.